# Confinement of unliganded EGFR by tetraspanin nanodomains gates EGFR ligand binding and signaling

Michael G. Sugiyama[1], Aidan I. Brown[2], Jesus Vega-Lugo[3], Jazlyn P. Borges [4], Andrew M. Scott[5], Khuloud Jaqaman [3,6], Gregory D. Fairn [7] & Costin N. Antonescu [1]✉

The epidermal growth factor receptor (EGFR) is a central regulator of cell physiology. EGFR is activated by ligand binding, triggering receptor dimerization, activation of kinase activity, and intracellular signaling. EGFR is transiently confined within various plasma membrane nanodomains, yet how this may contribute to regulation of EGFR ligand binding is poorly understood. To resolve how EGFR nanoscale compartmentalization gates ligand binding, we developed single-particle tracking methods to track the mobility of ligand-bound and total EGFR, in combination with modeling of EGFR ligand binding. In comparison to unliganded EGFR, ligand-bound EGFR is more confined and distinctly regulated by clathrin and tetraspanin nanodomains. Ligand binding to unliganded EGFR occurs preferentially in tetraspanin nanodomains, and disruption of tetraspanin nanodomains impairs EGFR ligand binding and alters the conformation of the receptor's ectodomain. We thus reveal a mechanism by which EGFR confinement within tetraspanin nanodomains regulates receptor signaling at the level of ligand binding.

The binding of epidermal growth factor (EGF) to its receptor (EGFR) triggers the activation of a wide range of intracellular signaling that in turn regulates many aspects of cell physiology[1–3]. EGFR is also a key driver of tumor progression and drug resistance in many cancers. Despite the importance of EGFR in physiology and disease, our understanding of the cellular mechanisms that control the initial stages of EGFR signaling such as ligand binding and receptor activation remain incomplete.

The binding of ligands such as EGF to EGFR in cells has complex kinetics reflecting multilayered regulation of this initial stage of EGFR signal activation. In addition to high (Kd of ~0.1 nM, ~10% of surface-exposed receptor) and low (Kd of 2–6 nM, ~90% of surface-exposed receptor) affinity subpopulations[4–9], the ligand binding affinity of EGFR is regulated by signaling, as very low concentrations of EGF reduce the ligand binding affinity of unliganded receptors[10]. Structural studies provide some key insights into the potential regulation of EGFR ligand binding. The EGFR ectodomain is comprised of four domains (I-IV), and EGF binding to domains I and III is coupled to a conformational change that allows formation of back-to-back EGFR dimers via contacts within domain II[11,12]. This conformational rearrangement of the ectodomain is coupled to formation of asymmetric dimers and phosphorylation of the EGFR C-terminal tail on multiple distinct tyrosine residues, each of which contribute to activation of specific signaling pathways[13].

EGFR extracellular domains II and IV form contacts leading to a closed or tethered conformation that disfavours activating

[1]Department of Chemistry and Biology, Toronto Metropolitan University, Toronto, ON, Canada. [2]Department of Physics, Toronto Metropolitan University, Toronto, ON, Canada. [3]Department of Biophysics, UT Southwestern Medical Center, Dallas, TX, USA. [4]Program in Neuroscience and Mental Health, Hospital for Sick Children, Toronto, ON, Canada. [5]Olivia Newton-John Cancer Research Institute, La Trobe University, Melbourne, VIC, Australia. [6]Lyda Hill Department of Bioinformatics, UT Southwestern Medical Center, Dallas, TX, USA. [7]Department of Pathology, Dalhousie University, Halifax, NS, Canada. ✉e-mail: cantonescu@torontomu.ca

dimerization; mutation of these residues increased ligand binding[14–16]. Molecular dynamics simulation and structural studies predict that the transition of EGFR from this tethered (closed) to the open ligand-bound state happens through an extended, open conformation of the EGFR ectodomain[14]. The complex regulation of EGFR ligand binding suggests that this transition from tethered (closed) to open (ligand-bound) conformations is not merely stochastic but controlled by

cellular mechanisms, yet the molecular underpinnings of this regulation remain poorly understood.

The study of distinct behaviours of individual EGFR molecules that may each exhibit distinct regulation of ligand binding is supported by observations made with single particle tracking (SPT) of EGFR to monitor receptor mobility, reviewed by ref. [17]. To date, SPT of EGFR has largely relied on fluorescent labeling of an anti-EGFR antibody[18,19],

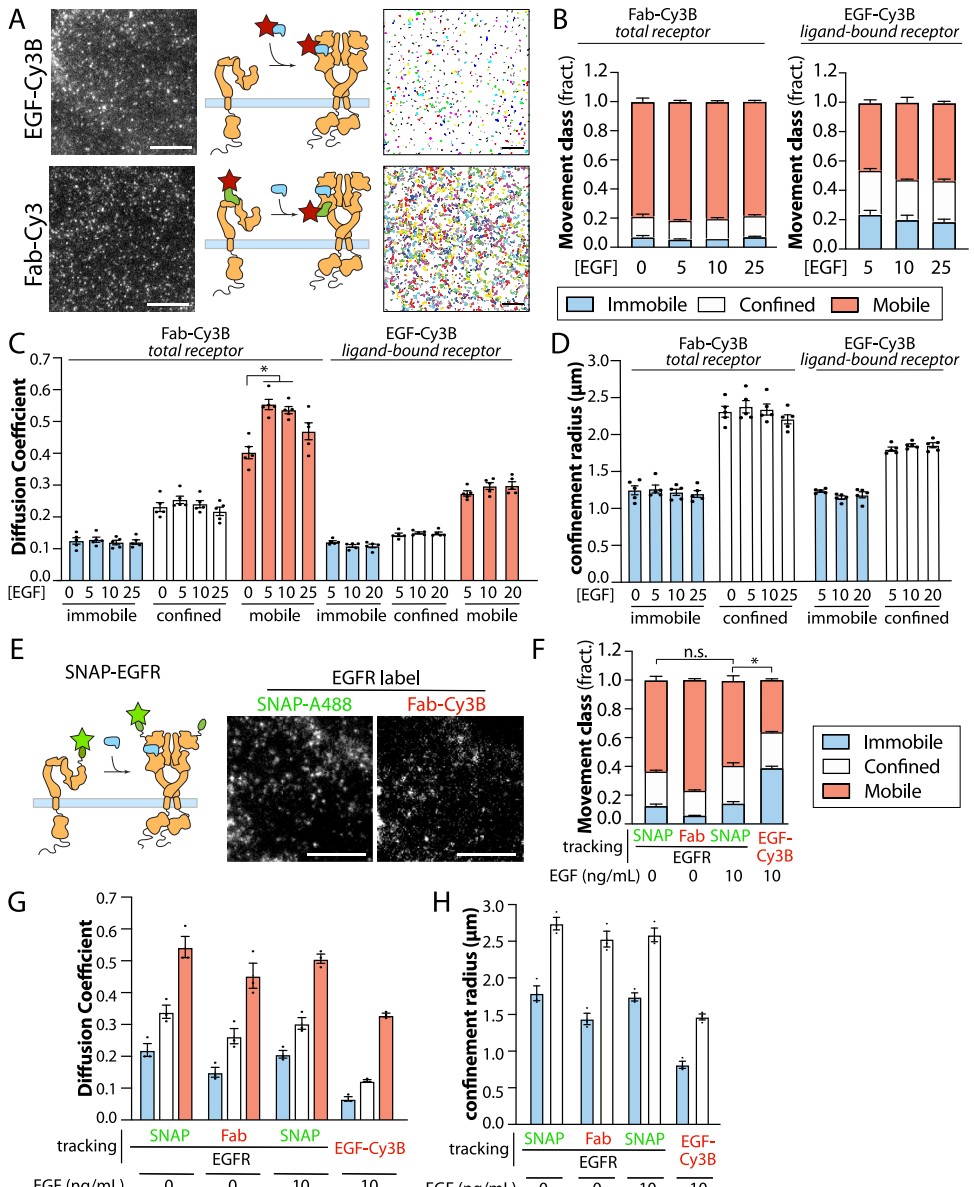

**Fig. 1 | SPT analysis reveals that ligand binding is gated and restricted to a subset of EGFR. A** Diagram showing labeling strategies for detection of ligand-bound receptors (via EGF-Cy3B, top panels) or all receptors, regardless of ligand binding status (via Fab-Cy3). Shown (left panels) are representative single-frames of time-lapse image series obtained by TIRF-M using each labeling strategy, as used for SPT analysis. Scale 3 μm. Also shown (right panels) are the results of the tracking showing traces of the EGFR particles over 25 frames (0.5 s). Scale 10 μm **B–D** Results of SPT analysis. **B** Shown are the mean ± SE of the fraction of all EGFR tracks, as labelled by Fab-Cy3B (left panels, "total receptor"), or the fraction of only ligand-bound EGFR, as labeling EGF-Cy3B (right panels, "ligand-bound receptor") that exhibit mobile, confined or immobile behaviour. Shown are mean ± SE of diffusion coefficient (**C**) or the confinement radius (**D**). EGF-Cy3B (ligand-bound) data is from 5 independent experiments, and Fab-Cy3B (total EGFR) data is from 4 independent experiments. Each experiment involved detection and tracking of >500 EGFR

objects. *$p < 0.05$. **E** Diagram showing SNAP-reagent EGFR labeling strategy that detects all receptors, regardless of ligand binding status. Also shown (right panels) are representative single-frames of time-lapse image series obtained by TIRF-M using each labeling strategy, as used for SPT analysis. Scale 3 μm. **F–H** Results of SPT analysis with SNAP-A488 (no EGF) and Fab-Cy3B performed immediately one after the other in the same cells, or SNAP-A488 (in the presence of 10 ng/mL EGF) and EGF-Cy3B (at 10 ng/mL) similarly performed sequentially in the same cells. **F** Shown are the mean ± SE of the fraction of all EGFR tracks that exhibit mobile, confined or immobile behaviour. Also shown are mean ± SE of diffusion coefficient (**G**) or the confinement radius (**H**). For **E–H**, the data is from 3 independent experiments. Each experiment involved detection and tracking of >500 EGFR objects. *$p < 0.05$. Statistical analysis and $p$-values are indicated in Supplementary Table 1. Source data are provided as a Source Data file.

Fab[20,21], or EGF ligand[18,22–26], or fluorescent protein fusion to EGFR[27–30]. These studies established that EGFR exhibits heterogeneity with respect to mobility, with each study reporting that specific fractions of the receptor are immobile and/or confined. The remaining EGFR exhibited mobile behaviour, and in some cases directed movement, such as along filopodia[23]. Extending these insights to resolve the regulation of EGFR confinement in relation to ligand-binding and signaling is currently an important priority.

Several SPT studies used broad perturbations, for example extraction of cholesterol or disruption of actin dynamics, finding that each impacts EGFR confinement[17]. Given the broad impact of these perturbations, the molecular mechanisms that establish the heterogenous confinement of EGFR remain poorly understood. Some studies have suggested that EGFR confinement could result from formation of EGFR oligomers, either in the basal state favouring EGFR inactivation[31] or in the ligand-bound state resulting in propagation of signaling beyond the activated EGFR dimer[32–34].

Association with other cell surface structures also controls EGFR confinement. CD82, which forms tetraspanin enriched nanodomains at the plasma membrane, contributes to EGFR confinement[19], as does clathrin[30], a well-known scaffold protein required for formation of clathrin-coated pits leading to receptor internalization[35,36]. These studies indicate that the confinement of EGFR may result from both receptor-intrinsic (e.g. homo-oligomerization) as well as receptor-extrinsic (association with tetraspanins or clathrin) factors. However, the role of specific nanodomains in regulating EGFR confinement and signaling at the level of EGFR ligand binding has only begun to be examined.

Signaling by EGFR and other receptors is regulated by nanoscale organization of receptors and downstream transducers/effectors into specific nanodomains at the plasma membrane[37]. This includes 200–400 nm nanodomains formed by assembly of tetraspanin proteins[37,38]. The tetraspanins CD82, CD81, CD9 interact with EGFR and together with CD151[39,40] may regulate EGFR signal transduction[41–46], by enrichment of signaling intermediates[47–51] within tetraspanin domains. Moreover, ligand binding by EGFR leads to enhanced recruitment to clathrin-coated pits and subsequent receptor endocytosis[35]. Clathrin-coated pits and endocytosis regulate more distal aspects of EGFR signaling such as Akt phosphorylation[52–54].

How engagement by EGFR in homo-oligomerization and/or recruitment to specific nanodomains each control receptor mobility and confinement, and how this impacts EGFR ectodomain conformation and ligand binding remains obscure. Here, we developed SPT methods to separately resolve the mobility of total and ligand-bound EGFR. This allowed us to establish a model in which the dynamic confinement within tetraspanin nanodomains imparts on a small subpopulation of EGFR enhanced ligand binding capacity. Upon EGF stimulation, ligand-bound EGFR is significantly more confined in a manner that is independent of tetraspanins and instead relies on both oligomerization and clathrin nanodomain confinement. We present a model whereby the tetraspanin resident EGFR is preferentially able to bind EGF, followed by rapid relocalization of ligand-bound EGFR out of tetraspanin nanodomains to other regions of the plasma membrane including clathrin nanodomains, supporting robust signal transduction.

## Results

### Distinct dynamics of ligand-bound and total EGFR

To resolve how ligand-bound and total EGFR may each exhibit heterogeneous properties and behaviour, we devised SPT approaches to separately study each category of receptor. To selectively label ligand-bound EGFR, we conjugated the EGF ligand directly to Cy3B (henceforth "ligand-bound EGFR", Fig. 1A). EGF-Cy3B elicited signaling indistinguishable from that triggered by unlabelled EGF (Supplementary Fig 1C), indicating that fluorescence conjugation did not alter

ligand binding. To selectively label total EGFR (regardless of ligand binding), we developed a Fab fragment from mAb108[55], which we conjugated to Cy3B (Fig. 1A). Importantly, while full-length mAb108 reduces binding of EGF to EGFR[55], saturating concentration of the Fab fragment did not alter EGF-Cy3B binding (Supplementary Fig. 1A, B). This indicates that this Fab does not reduce EGFR ligand binding and can thus act as a reliable reporter of total EGFR (whether ligand-bound or not, henceforth "total EGFR"). These parallel labeling strategies for total EGFR (via Fab-Cy3B) and ligand-bound EGFR (via EGF-Cy3B) provide strong signal-to-noise for imaging in TIRF-M at 20 Hz (Fig. 1A), ideal for SPT analyses[56].

Using this approach, we found that a large majority of (total) EGFR detected by Fab-Cy3B undergoes free diffusion (referred to as "mobile" throughout this work), as classified by moment scaling spectrum analysis[57,58] (Fig. 1B, left panel). The proportion of confined/immobile EGFR versus mobile EGFR changes significantly but only modestly upon addition of EGF at the highest concentration of [EGF] tested (200 ng/mL or 33 nM, Supplementary Fig. 1E). In contrast, EGF-bound EGFR (EGF-Cy3B) exhibits much more prominent confined and immobile fractions (Fig. 1B, right panel) relative to what was seen with total EGFR. We confirmed the validity of this SPT approach, as samples subject to fixation exhibited nearly complete confinement (Supplementary Fig. 1D).

To further validate that our Fab-based labeling strategy to detect total EGFR was minimally perturbing to EGFR behaviour, we developed a complementary receptor labeling strategy using the doxycycline-inducible Sleeping Beauty transposon system[59] to generate ARPE-19 cells that stably express EGFR fused to an N-terminal SNAP-tag. Using this system, we expressed the N-SNAP-EGFR at near endogenous levels (through control of [doxycycline]) (Supplementary Fig. 1F) and selectively label surface EGFR using a cell-impermeant SNAP reagent conjugated with the Atto488 fluorophore (Fig. 1E). Importantly, this strategy allows for the dual labeling of EGFR in the same cell using two separate labeling strategies and fluorophores (i.e., Fab/EGF-Cy3B and N-SNAP-EGFR-488, Fig. 1E). We acquired sequential time-lapses of Fab/EGF-Cy3B and N-SNAP-EGFR-488 in the same cells. Our results show that cells labeled with Fab-Cy3B and N-SNAP-EGFR-488 exhibit similar patterns of EGFR mobility (i.e., the vast majority of EGFR is mobile); in contrast, there is a stark difference in EGFR mobility when it is tracked in the same cell labeled with N-SNAP-EGFR-488 (highly mobile) and EGF-Cy3B (highly confined) (Fig. 1F). Together, these results support our approach for tracking total and ligand-bound EGFR under minimally perturbing conditions.

We also determined the diffusion coefficient (Fig. 1C, G) and confinement radius (Figs. 1D, H) of total EGFR in either the basal or EGF-stimulated states (detected by Fab-Cy3B or SNAP-A488) and ligand-bound EGFR (detected by EGF-Cy3B). As expected, the mobile class of EGFR had a higher diffusion coefficient than the confined and immobile classes, for both total EGFR and ligand-bound EGFR. While EGF stimulation did not impact the proportion of total EGFR in the mobile class (detected via Fab-Cy3B: Fig. 1B or SNAP-EGFR: Fig. 1E), EGF stimulation slightly increased the diffusion coefficient of the mobile cohort of total EGFR detected by Fab-Cy3B (Fig. 1C) but this minor effect was not recapitulated in the SNAP-EGFR cells (Fig. 1G). At the same time, the diffusion coefficient of the mobile cohort of ligand-bound EGFR was substantially lower than that of the mobile fraction of total EGFR (detected via Fab-Cy3B: Fig. 1C or SNAP-EGFR: Fig. 1G). Together, these results indicate that EGFR may participate in small-scale interactions that modulate the diffusion coefficient of mobile EGFR in both ligand-bound or non-ligand-bound states. For example, the reduced diffusion coefficient of the mobile fraction of ligand-bound EGFR (detected via Cy3B-EGF) compared to total EGFR (the majority of which are non-ligand bound even in the presence of EGF in these conditions) may reflect EGFR dimerization in the ligand-bound state or other similar small-scale interactions. Importantly, these

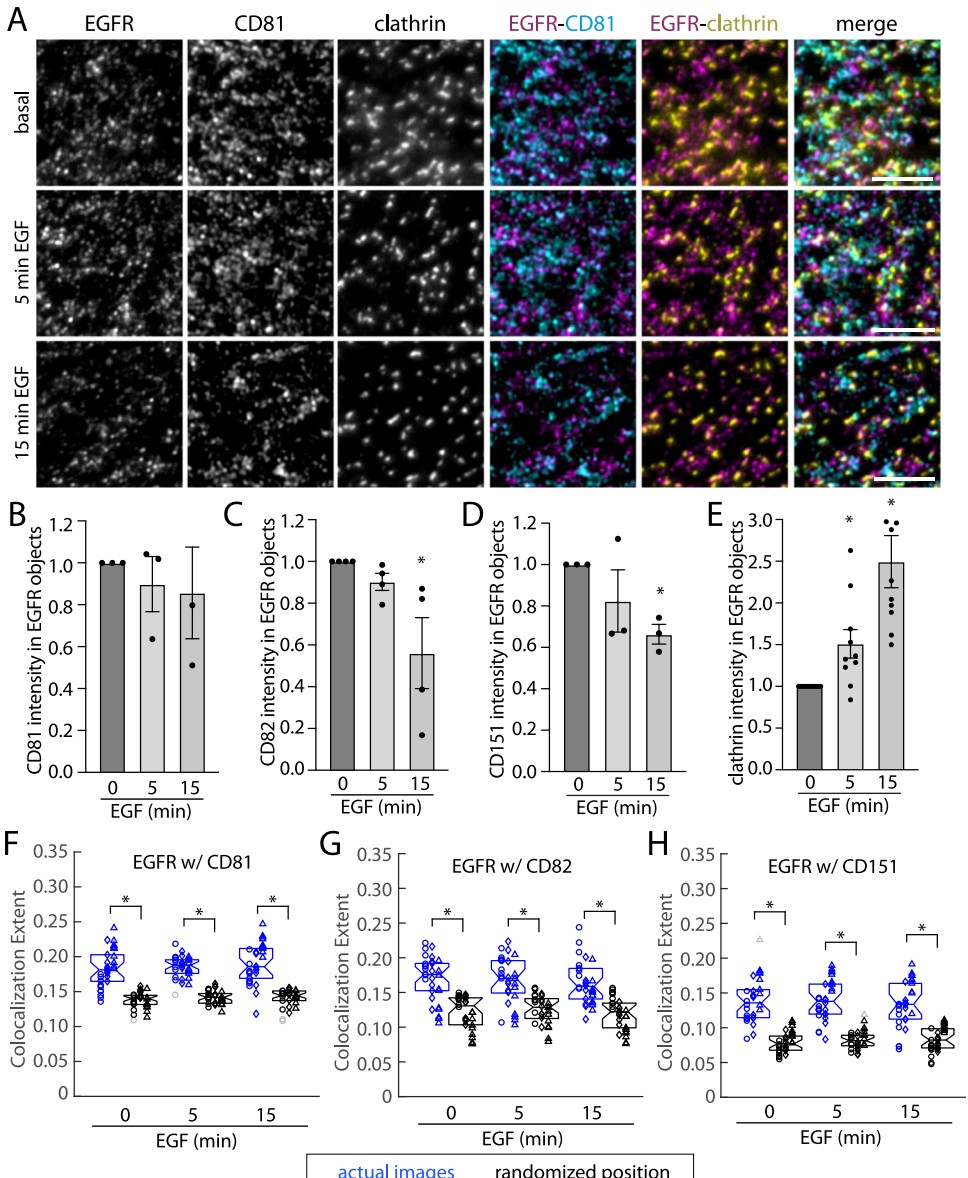

**Fig. 2 | EGF stimulation causes a redistribution of bulk EGFR from tetraspanin nanodomains to clathrin nanodomains.** ARPE-19 cells stably expressing eGFP-clathrin were labeled with Fab-Cy3B (to label total EGFR) and stimulated with EGF as indicated, followed by fixation and staining with CD81 antibodies. **A** Shown are representative images obtained by TIRF-M; antibody labeling of tetraspanins is highly specific (Supplementary Fig. 2); similar experiments with labeling of CD82 and CD151 were performed (Supplementary Fig. 4). Scale = 5 μm. **B**–**E** Shown are results of detection of EGFR objects followed by *intensity-based analysis* of EGFR object overlap with the indicated secondary channel (tetraspanins or clathrin) following subtraction of background (as described in Methods) with CD81, CD82, CD151, or clathrin. $n > 3$ independent experiments each with >10 cells and >5000 EGFR objects quantified. Shown are the means of marker intensity within EGFR objects in individual experiment (dots), and the overall mean ± SE; $^*p < 0.05$ relative to basal condition. **F**–**H** Shown are results showing the overlap of EGFR and tetraspanin objects followed by *position-based analysis* of EGFR and secondary markers, as described in Methods. Shown are the individual measurements per cell, grouped by individual experiments (circle, triangle, diamond), and experimental mean for actual image pairs (blue) and EGFR relative to randomized position of the secondary channel (black). $^*p < 0.05$. Statistical analysis and $p$-values are indicated in Supplementary Table 1. Source data are provided as a Source Data file.

results show that ligand-bound EGFR is substantially more confined/immobile than total EGFR.

As the fraction of confined or immobile EGFR does not change substantially upon EGF stimulation (Figs. 1B, F), these results suggest that only a subset of total EGFR can bind EGF, even at very high levels of [EGF], as ligand-bound EGFR is significantly more confined or immobile. Based on Kd values for EGFR ligand binding, approximately 10–40% of EGFR binding sites are expected to be occupied with ligand at physiological levels of EGF (5–10 ng/mL) (Supplementary Fig. 1G); however, EGFR binding is complex[4–9], and higher EGF concentrations may result in binding that deviates from that expected from a single Kd. As such, we focused on the study of ligand binding at these

physiological concentrations (5–10 ng/mL). At these conditions (5–10 ng/mL EGF), we observe that selective tracking of ligand-bound EGFR (via EGF-Cy3B) yields mostly confined or immobile EGFR, yet we do not observe a robust ligand-induced increase in EGFR confinement of total EGFR at 5–10 ng/mL EGF (compared to the 0 ng/mL condition, Fig. 1B).

This suggest two possibilities: (i) that ligand binding occurs selectively in EGFR that is pre-confined in the basal state, with continued confinement of ligand-bound EGFR by the same mechanism, or (ii) that ligand binding to EGFR triggers confinement that is distinct from that of unliganded EGFR, yet is concomitant to loss of confinement of pre-confined EGFR. Hence, we next sought to define

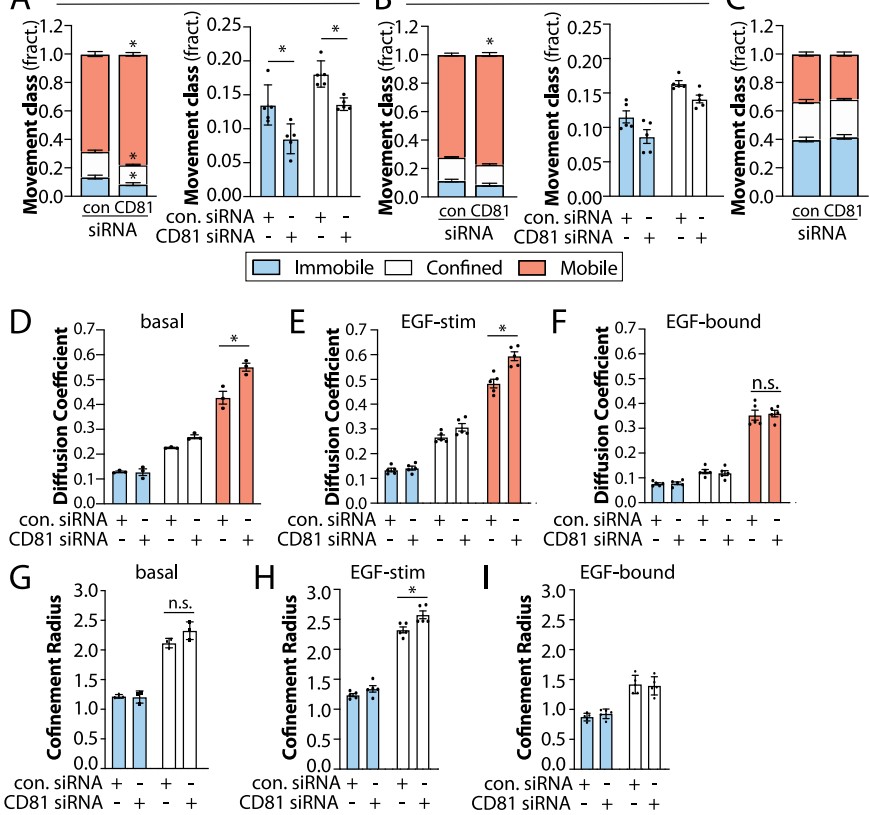

**Fig. 3 | A subset of EGFR is confined within tetraspanin nanodomains and exhibits preferential EGF binding.** ARPE-19 cells were treated with siRNA to silence CD81, or non-targeting siRNA (control), as indicated. **A**–**I** Results of SPT analysis. The cells were subjected to SPT using either Fab-Cy3B to label total EGFR in the absence or presence of unlabelled EGF, or labelled using EGF-Cy3B to label only ligand-bound EGFR. Shown in **A**–**C** are the mean ± SE of the fraction of EGFR tracks in each mobility category (immobile, confined, mobile) under each condition. In each of the panels **A**, **B**, we replot the same data to show only the immobile and confined fractions. Also shown are mean ± SE of diffusion coefficient (**D**–**F**) or the confinement radius (**G**–**I**). EGF-Cy3B (ligand-bound) data is from 4 independent experiments, and Fab-Cy3B (total EGFR) data is from 5 independent experiments. Each experiment involved detection and tracking of >500 EGFR objects. $^*p < 0.05$. Statistical analysis and $p$-values are indicated in Supplementary Table 1. Source data are provided as a Source Data file.

how specific protein-driven nanodomains contribute to EGFR confinement in the basal and EGF-stimulated states, focusing on the effect of 10 ng/mL EGF treatment.

## Systematic analysis of nanodomain enrichment of EGFR at the cell surface

The distribution of EGFR within various plasma membrane nanodomains has been examined previously (reviewed in refs. [37,60]). However, few studies have studied EGFR enrichment in more than one candidate nanodomain at a time and in the same cell model used to study EGFR mobility by SPT, making conclusions about the relationship of confinement and nanodomain enrichment challenging. Hence, we used labeling strategies that allowed detection of total EGFR (via Fab-Cy3B) alongside simultaneous labeling of multiple nanodomain compartment markers, such as tetraspanins, using specific antibodies (Supplementary Fig. 2B).

This imaging was coupled to automated detection of EGFR via a Gaussian-based modeling approach[52,61–63], which allows measurement of the levels of a particular marker (EGFR or secondary 'nanodomain' channels) at the position of the detected EGFR object. To determine specific detection of secondary 'nanodomain' channels within EGFR objects, background overlap was determined by repeating measurements after the position of the EGFR image had been randomized. This approach revealed that caveolin-1 and flotillin-1 are not significantly detected in EGFR objects under either basal or EGF-stimulated conditions (Supplementary Fig. 3A, F, G). In contrast, each of the tetraspanins CD81 (Fig. 2), CD82 (Supplementary Fig. 4A) and CD151

(Supplementary Fig. 4B) as well as clathrin (Fig. 2A, Supplementary Fig. 3, 4) were significantly detected within EGFR objects in the basal and EGF-stimulated condition (Supplementary Fig. 3B–E).

Using this approach and subtracting background (random) overlap allowed assessment of the effect of EGF stimulation on EGFR overlap with tetraspanin and clathrin plasma membrane nanodomains (Fig. 2). EGF did not impact the level of CD81 associated with EGFR objects (Fig. 2B) or the total abundance of CD81 on the cell surface (Supplementary Fig. 3H) but it reduced the level of CD82 (Fig. 2C) or CD151 (Fig. 2D) associated with EGFR. In contrast, EGF stimulation robustly increased the level of clathrin detected within EGFR objects (Fig. 2E). As this approach reveals the average levels of tetraspanins or clathrin associated with EGFR objects, these results could either indicate that upon EGF stimulation, individual EGFR objects associate with smaller-scale assemblies of CD82 and CD151 (e.g., nanodomains to small oligomers/clusters) or that a subset of EGFR disengages completely from tetraspanin association. To resolve this, we used a complementary method of analysis that determines the probability of overlap of EGFR with a secondary marker, considering only the position of EGFR and secondary channel objects but not their intensities[58,64]. This analysis showed significant association of EGFR with each of CD81 (Fig. 2F), CD82 (Fig. 2G) and CD151 (Fig. 2H), and notably this overlap of EGFR and each tetraspanin was not affected by EGF stimulation.

These results imply that there is no detectable change in the fraction of EGFR associated with tetraspanins upon EGF stimulation, but that EGF stimulation triggers a decrease in the size of tetraspanin

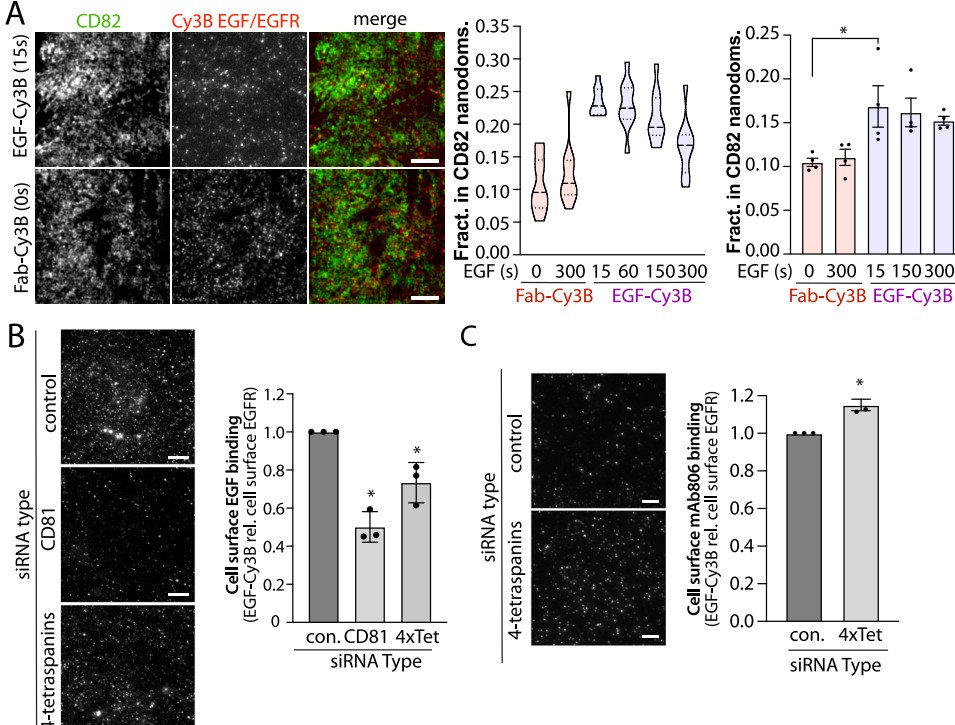

**Fig. 4 | Tetraspanin nanodomains regulate EGFR ligand binding. A** ARPE-19 cells were either (i) pre-labelled with Fab-Cy3B (to label total EGFR), and then treated with unlabelled EGF as indicated, or (ii) labelled with EGF-Cy3B for short times (as low as 15 s) as indicated. After this labeling, cells were fixed, stained for CD82, and subject to TIRF-M imaging and analysis as described in Methods. Shown are representative images (left panel), and the measurement of the fraction of Cy3B puncta within CD82 tetraspanin nanodomains in an individual experiment showing the distribution of measurements in individual cells (middle panel). Results are shown as the distribution of measurements in individual cells (violin plot), featuring the 25th, 50th and 75th percentiles (horizonal dashed lines). Also shown (right panel) are the mean of these values from $n = 4$ individual experiments (dots) $\pm$ SE; $^*p < 0.05$ (right panel). Scale 10 μm (**B**) ARPE-19 cells were transfected with siRNA targeting CD81 or 4 tetraspanins simultaneously (4x-tet: CD81, CD82, CD151 and CD9), then labelled with Cy3B-EGF for 2 min at 37 °C, followed by rapid

washing and fixation. Shown are representative images of cell surface Cy3B-EGF labeling (left panels), scale 20 μm, as well as the mean intensity of cell surface EGF-Cy3B labeling per cell, normalized to cell surface EGFR levels in each condition (Supplementary Fig. 10). Shown are the overall mean (bars) and the mean cellular values from individual experiments (dots) $\pm$SE; $^*p < 0.05$ relative to control siRNA condition. **C** Intact (non-permeabilized) cells were incubated with mAb806 at 37 °C for 5 min, followed by fixation and washing of unbound antibodies. Shown are representative images obtained by TIRF microscopy, as well as quantification of mAb806 binding, normalized to cell surface EGFR levels in each condition (Supplementary Fig. 10). Shown are the overall mean (bars) and the mean cellular values from individual experiments (dots) $\pm$SE; $^*p < 0.05$ relative to control siRNA condition. Scale 20 μm. For **B**, **C**, the data is from 3 independent experiments. Statistical analysis and $p$-values are indicated in Supplementary Table 1. Source data are provided as a Source Data file.

assemblies associated with EGFR. This in turn suggests that there are multiple scales of EGFR association with tetraspanins–with larger scale assemblies, perhaps tetraspanin nanodomains preferred in the basal state and smaller scale assemblies that are retained upon EGF stimulation. To further examine this possibility, we performed additional analyses to determine the extent to which tetraspanins form different-size assemblies on the plasma membrane. Automated detection of diffraction-limited CD81 objects in TIRF images revealed that these objects have a range of intensities, consistent with of the formation of CD81 assemblies of varying sizes (Supplementary Fig. 3I). Then, we subdivided these CD81 structures into two groups representing small and large assemblies based on an arbitrary but systematic intensity threshold to determine if EGFR preferentially associates with one or the other. This analysis showed significant association of EGFR with both small and large CD81 assemblies relative to the randomized images (Supplementary Fig. 3J). We also performed super-resolution STED microscopy on ARPE-19 cells to image endogenous CD81, which revealed large tetraspanin assemblies consistent with tetraspanin nanodomains, as well as other smaller-scale objects (Supplementary Fig. 3K). Considering our results from the intensity-based (Fig. 2B–E) and position-based (Fig. 2F–H) analyses, the current size-based analysis and super-resolution imaging of tetraspanins supports the notion that tetraspanins form multiple levels of assembly that are capable of associating with EGFR. EGFR association with larger assemblies of

tetraspanins (that could be tetraspanin nanodomains) may result in outright confinement/immobilization, while EGFR association with smaller assemblies of tetraspanins may modulate the diffusion of the mobile fraction.

We observed that EGF-stimulation induces robust overlap with clathrin (Fig. 2E), reflecting recruitment of EGFR to clathrin-coated pits, which are large nanodomains expected to confine or immobilize EGFR. This interpretation of an apparent shift of enrichment of a subset of EGFR from one nanodomain comprised of larger-scale tetraspanin assemblies to another larger-scale nanodomain (clathrin) upon EGF stimulation is consistent with our observation that EGF stimulation does not elicit a robust change in the overall fraction of EGFR in each of the mobility classes. This suggests a model in which prior to ligand-binding, a subset of EGFR is pre-confined within tetraspanin nanodomains, and that following ligand binding, ligand-bound EGFR disengages from large tetraspanin assemblies while retaining small-order tetraspanin association, and is rapidly recruited and confined by another mechanism, which may include clathrin nanodomains.

### Distinct confinement requirements for non-ligand bound versus ligand-bound EGFR

The association of EGFR with small-scale tetraspanins assemblies may reveal a role for tetraspanins in regulating the diffusion of mobile EGFR. In contrast, EGFR association with larger-scale tetraspanin

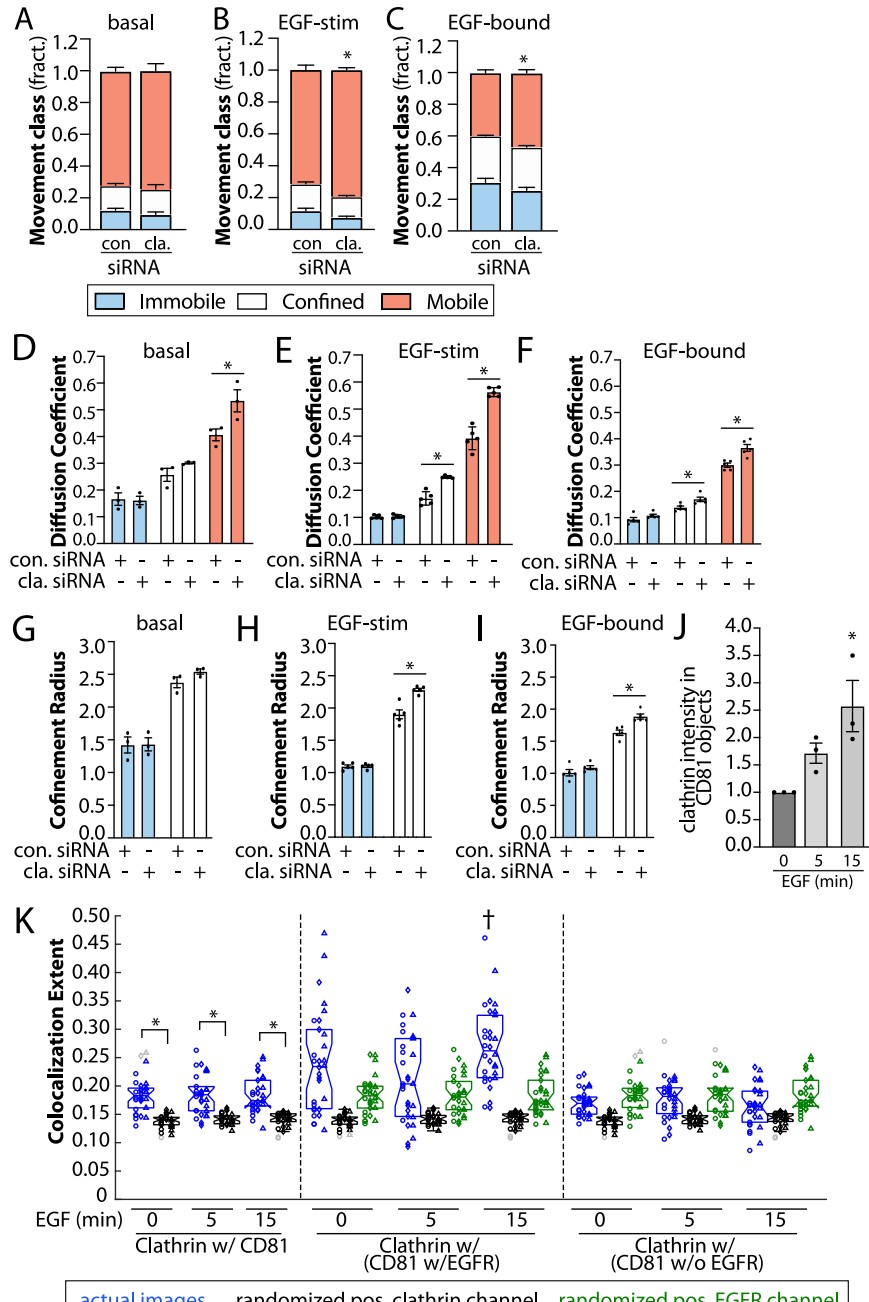

**Fig. 5 | Clathrin silencing preferentially impacts confinement of ligand-bound EGFR.** APRE-19 cells were treated with siRNAs targeting clathrin heavy chain (clathrin) or non-targeting siRNA (con.). Cells were subjected to SPT using either Fab-Cy3B to label total EGFR in the absence (**A, D, G**) or presence (**B, E, H**) of unlabelled EGF, or labelled using EGF-Cy3B (**C, F, I**). A–C are the mean ± SE of the fraction of EGFR tracks in each mobility category. Also shown are mean ± SE of diffusion coefficient (**D–F**) or confinement radius (**G–I**). Data is from 4 and 5 independent experiments for EGF-Cy3B (ligand-bound) Fab-Cy3B (total EGFR), respectively. Each experiment involved detection and tracking of >500 EGFR objects. *$p < 0.05$. **J** An additional analysis of the images from Fig. 2 involving detection of CD81 objects followed by *intensity-based analysis* of overlap with eGFP-clathrin, following subtraction of background (as described in Methods) n = 3 independent experiments each with >10 cells and >5000 EGFR objects quantified. Shown are the means of eGFP clathrin intensity within CD81 objects in individual experiment (dots), and the overall mean ± SE; *$p < 0.05$ relative to basal condition. **K** An

additional analysis of the images shown in Fig. 2. Results of conditional, *position-based analysis*, as in Methods. Shown are the individual measurements per cell of clathrin colocalization with CD81 (regardless of EGFR context, left panels), clathrin colocalization with CD81 (conditional on additional co-localization of EGFR objects, middle panels, shaded grey), clathrin colocalization with CD81 (conditional on exclusion from co-localization of EGFR objects, right panels). Individual cellular values are grouped by experiments (circle, triangle, diamond), and experimental mean for actual image sets (blue) as well as that from randomized position of clathrin (black) and EGFR (green) channel objects relative to CD81 objects. *$p < 0.05$, †$p < 0.5$ relative to corresponding colocalization extent in (i) clathrin w/ CD81 (irrespective of EGFR), (ii) clathrin w/ (CD81 w/EGFR) with randomized clathrin positions and (iii) clathrin w/ (CD81 w/EGFR) with randomized EGFR positions. Statistical analysis and *p*-values are indicated in Supplementary Table 1. Source data are provided as a Source Data file.

assemblies (e.g. in nanodomains) could impact the fraction of EGFR that is confined or immobile. Our SPT methods are thus ideally suited to resolve how tetraspanins may distinctly modulate the mobility of ligand-bound and non-liganded EGFR.

We examined the effects of silencing of several tetraspanins reported to have some interaction with EGFR or with one another (Supplementary Fig. 2A, B), on the mobility of EGFR as assessed by SPT. Silencing of CD81 significantly reduced the fraction of Fab-Cy3B tracks classified as immobile or confined, while increasing the fraction of mobile tracks in the basal state (Fig. 3A). Silencing of CD81 appeared to have similar yet more modest effect on Fab-Cy3B tracks in the EGF-stimulated state (Fig. 3B) and was without effect on the fraction of EGFR in each mobility cohort of ligand-bound EGFR (Fig. 3C). Similar results were obtained in MDA-MB-231 cells (Supplementary Fig. 5). Consistent with our localization analyses (Fig. 2), this suggests that CD81 plays an important role in the confinement or immobilization of unliganded EGFR, which together with localization of EGFR to tetraspanin nanodomains in the basal state (Fig. 2) may reflect recruitment of this cohort of EGFR into larger-scale tetraspanin assemblies prior to ligand binding. Because tetraspanin nanodomains are 100–200 nm in diameter, the confinement radius of immobile and confined receptor classes of 1–2.5 μm likely reflects technical limitations related to imaging resolution, labeling strategy, and/or classification of individual EGFR tracks as entirely immobile, confined or mobile, as receptors may exhibit switching of behaviours during the observation period at rates below the current limit of detection. Nonetheless, these results indicate that CD81 controls the fraction of EGFR exhibiting confined/immobile behaviour selectively in the unliganded state, suggesting that EGFR is confined in tetraspanin nanodomains prior to ligand binding.

Interestingly, silencing CD81 also resulted in an increase in the diffusion coefficient of the mobile fraction of total EGFR (detected via Fab-Cy3B), both in the basal (Fig. 3D) and in the EGF-stimulated condition (Fig. 3E), but not for the ligand-bound EGFR (Fig. 3F). Silencing of CD82, CD9 or CDC151 did not alter the proportion of EGFR within the mobility classes (Supplementary Fig. 6A–C, 7A–C, 8A–C). However, silencing of each of CD82, CD9 and CD151 increased the diffusion coefficient of the mobile fraction, both for unliganded EGFR and ligand-bound EGFR (Supplementary Fig. 6D–F, 8D–F). This indicates that in addition to a specific role for CD81 in establishing a subset of confined/immobile EGFR in the unliganded state, all tetraspanins tested contribute in some capacity to decreasing the diffusion of mobile EGFR. This is consistent with the observation that EGFR remains associated with small-scale tetraspanin assemblies upon EGF stimulation (Fig. 2). Concomitant silencing of all four of these tetraspanins was also without effect on the mobility of ligand-bound EGFR, detected via EGF-Cy3B tracking (Supplementary Fig. 9A), indicating that the lack of effect of silencing of each tetraspanin on the confinement of ligand-bound EGFR was unlikely the result of redundancy among these four tetraspanins. As tetraspanin silencing did not impact the fraction of confined or immobile ligand-bound EGFR, confinement of EGFR in larger-scale tetraspanin assemblies (tetraspanin nanodomains) may not be compatible with ligand-bound EGFR.

These results support a model in which the cohort of non-ligand-bound EGFR are pre-confined in a CD81-dependent manner within tetraspanin nanodomains that may contain CD81, CD82, CD9 and CD151, where EGFR may exhibit preferential association with EGF. We thus next examined if EGFR within tetraspanin nanodomains may exhibit preferential binding to EGF. To do so, we examined if short times of incubation with EGF-Cy3B would result in preferential labeling within tetraspanin domains compared to labeling by Fab-Cy3B (total EGFR). Following labeling with Fab-Cy3B or EGF-Cy3B, labeling of CD82, imaging by TIRF-M and detection of Cy3B puncta, we used an arbitrary but systematic threshold to define CD82-positive Cy3B puncta. Stimulation for 15 s resulted in EGF-Cy3B puncta that were

significantly enriched within CD82 structures compared to Fab-Cy3B puncta (Fig. 4A). This experiment indicates that EGF exhibits preferential, but perhaps not exclusive, association with EGFR within tetraspanin nanodomains relative to the entire pool of total EGFR within the plasma membrane.

The preferential association of EGF with EGFR detected alongside tetraspanins suggests that tetraspanins may regulate EGFR ligand binding. Silencing CD81 alone or all four tetraspanins in combination significantly reduced EGF-Cy3B cell surface binding (Fig. 4B) relative to cell surface EGFR levels (Supplementary Fig. 10), suggesting that tetraspanins might regulate ligand binding for a fraction of EGFR at the earliest detectable stages. mAb806 is a conformationally-sensitive EGFR antibody that does not bind to tethered, closed EGFR or ligand-bound conformation[14,65–67]. Thus, a change in mAb806 binding to EGFR may reflect a change in conformation and/or in protein-protein interactions of the EGFR ectodomain, although the extent of mAb806 binding may not be directly proportional to ligand affinity. Tetraspanin silencing elicited a small but significant increase in the cell-surface binding of mAb806 relative to cell surface EGFR levels (Fig. 4C), Hence, tetraspanins associate with EGFR at different scales of assembly, including outright confinement that is selective for the non-ligand bound EGFR, and tetraspanins regulate EGFR ectodomain conformation and/or protein-protein interactions, which are associated with enhanced EGFR ligand binding.

As EGF stimulated enrichment of EGFR in clathrin-labelled structures (Fig. 2A, E), we next examined if the confinement of ligand-bound EGFR was functionally dependent on clathrin heavy chain (henceforth clathrin), the core protein for formation of clathrin-coated pits. Silencing clathrin did not impact the proportion of total EGFR (detected via Fab-Cy3) in each mobility class in the absence of EGF ligand (Fig. 5A). In contrast, clathrin silencing increased the proportion of total EGFR exhibiting mobile behaviour in the EGF-stimulated state (Fig. 5B) as well as that of ligand-bound EGFR (detected via EGF-Cy3B) (Fig. 5C).

In addition, clathrin silencing resulted in a significant increase in the diffusion coefficient of total EGFR in the absence (Fig. 5D) or presence (Fig. 5E) of EGF stimulation, and of ligand-bound EGFR (Fig. 5F), as well as a modest but significant increase in the confinement radius of confined but not immobile EGFR, selectively in the presence of EGF stimulation (Fig. 5G–I). Clathrin silencing impairs the formation of clathrin-coated pits that recruit and immobilize/confine ligand-bound EGFR prior to their eventual endocytosis, but clathrin silencing may also indirectly affect EGFR mobility due to broad impairment of endocytosis. To resolve if the decrease in the fraction of immobile/confined of ligand-bound EGFR was due to indirect, broad effects resulting from loss of endocytosis, we examined the effect of dynamin2 silencing. Dynamin2 facilitates the final stage of vesicle formation by catalyzing scission of vesicles from the plasma membrane. Silencing dynamin2 impairs endocytosis but allows formation of clathrin assemblies at the cell surface and recruitment of EGFR therein[52]. Silencing of dynamin2 did not affect the mobility of total or ligand-bound EGFR (Supplementary Fig. 9B), suggesting that the changes we observe in EGFR dynamics following clathrin heavy chain silencing are not due to a broad defect in endocytosis. Hence, the decrease in the confined fraction of EGFR upon clathrin silencing likely reflects loss of clathrin-coated pits capable of capturing ligand-bound EGFR in this condition. The increase in diffusion coefficient and confinement in various mobility cohorts upon clathrin silencing (Fig. 5B–F, Supplementary Fig. 9) is consistent with clathrin also forming highly transient assemblies that may undergo rapid abortive turnover without formation of bona fide clathrin-coated pits[68], which become further de-stabilized under clathrin silencing conditions where clathrin abundance is limiting. These abortive clathrin assemblies that are transient

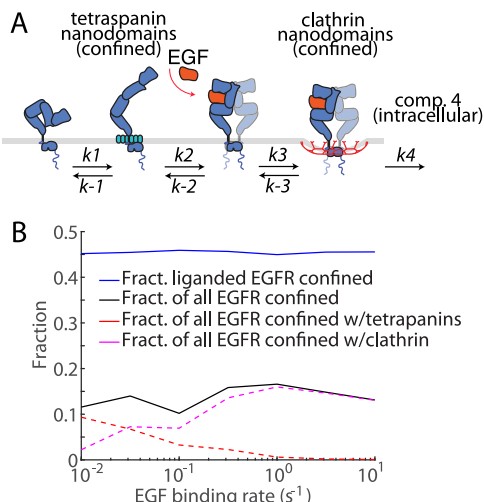

**Fig. 6 | Model of EGFR compartment transitions. A** EGFR (blue) compartmentalization and ligand-binding model. Unliganded EGFR outside of nanodomains cannot become ligand-bound. EGFR can transition into a tetraspanin domain (turquoise), where EGF (orange) binding is permitted. Unliganded EGFR slowly exit a tetraspanin domain, while ligand-bound EGFR rapidly depart a tetraspanin domain and can be confined to a clathrin domain and removed from the membrane. The $k_i$ describe the rates of transition between compartments. **B** shows the confined EGFR fractions from a stochastic quantitative model of EGFR diffusion and ligand binding dynamics on the membrane, following (**A**). The black solid line represents the fraction of all EGFR that are confined to nanodomains, comparable to immobile+confined fraction in the left panel of Fig. 1B. The blue solid line represents the fraction of liganded EGFR confined to nanodomains, comparable to the immobile+confined fraction in the right panel of Fig. 1B. Red and magenta dashed lines represent the fraction of all EGFR confined to tetraspanin and clathrin nanodomains, respectively, comparable to the control data in Fig. 2C. This data is for slow unliganded EGFR entry and exit from tetraspanin nanodomains. Each data point is averaged over 100 EGFR trajectories. Statistical analysis and p-values are indicated in Supplementary Table 1. Source data are provided as a Source Data file.

(some with lifetimes <3–5 s)[68] are expected to impact the diffusion of mobile EGFR without leading to outright receptor confinement or immobilisation that occurs within bona fide clathrin-coated pits and precedes endocytosis. Nonetheless, while the high sensitivity of these SPT measurements reveals unprecedented insight into the role of clathrin in transient association with mobile EGFR, these results reflect a preferential role for clathrin in recruitment of EGFR to immobile and confined receptor fractions when ligand-bound.

Our observations that ligand-bound EGFR remains associated with small-scale tetraspanin assemblies predicts that EGF stimulation would result in increased association of tetraspanins with clathrin, given the recruitment of EGFR to clathrin structures upon EGFR stimulation (Fig. 2). To investigate this, we focused on CD81 and performed additional analysis on the images presented in Fig. 2 to detect the overlap of CD81 objects with clathrin. As predicted, EGF stimulation triggered an increase in the intensity of clathrin within CD81 puncta (Fig. 5J). To determine if the increase of clathrin associated with CD81 objects correlated with EGFR recruitment to these same structures, we analyzed these images using conditional colocalization analysis[58]. This analysis allowed us to determine whether the probability of colocalization between CD81 and clathrin puncta increased when they also colocalized with EGFR. Like the colocalization analysis in Fig. 2F–H, this analysis was irrespective of object intensity. Using this approach, the probability of overlap of clathrin and CD81 puncta was found to be higher in the condition of additional overlap with EGFR than in its absence (Fig. 5K). Hence, consistent with the association of EGFR with small-scale assembly CD81 and other tetraspanins, these results indicate

that CD81 and EGFR exhibit co-recruitment into clathrin-coated pits upon EGF stimulation.

Together, our results indicate that upon stimulation at 10 ng/mL, EGF binding occurs preferentially for EGFR pre-confined within tetraspanin nanodomains, and that ligand binding triggers the rapid loss of EGFR from tetraspanin confinement, and at the same time confinement of ligand-bound EGFR by other mechanisms that include EGFR recruitment to clathrin-coated pits.

### Quantitative modeling recapitulates EGFR nanodomain confinement measures

Our results suggest larger-scale tetraspanin domains can confine unliganded EGFR and promote EGF binding, and that once ligand-bound, EGFR has reduced dependence on tetraspanins for confinement and instead can be confined by clathrin nanodomains. This sequence of EGFR states is described by a model with four distinct states in Fig. 6A: unliganded EGFR can enter and exit larger-scale tetraspanin assemblies (tetraspanin nanodomains) where the receptor becomes confined; ligand binding occurs preferentially to unliganded EGFR confined in tetraspanin domains, allowing EGFR exit from these nanodomains; ligand-bound EGFR can be confined by clathrin domains and/or oligomerization; and ligand-bound EGFR in clathrin domains can be removed from the cell membrane (e.g. via clathrin endocytosis).

To further explore EGFR dynamics, we developed a spatial, stochastic, agent-based computational model, consistent with the description of Fig. 6A. It includes EGFR diffusion on the membrane, ligand binding, entry into and exit from tetraspanin and clathrin domains, and removal from the membrane. The model simulates individual EGFR as they diffusively explore a membrane region containing both tetraspanin and clathrin domains. Energy barriers impede unliganded EGFR entry and exit from tetraspanin domains, and ligand-bound EGFR entry and exit from clathrin domains. In this model, EGFR may only become ligand bound inside tetraspanin domains, and we only consider EGFR association within larger-scale tetraspanin assemblies that results in confinement, but not association of EGFR with smaller-scale tetraspanin assemblies. Further, in this model, the energy barriers confining unliganded EGFR to tetraspanin nanodomains are immediately (0 s) removed after ligand binding, facilitating rapid diffusive exit of liganded EGFR from tetraspanin nanodomains, an approximation supported by the lack of effect of tetraspanin silencing on the fraction of confined/immobile EGFR in the ligand-bound state (Fig. 3C).

This model determines the fraction of EGFR confined within each type of nanodomain at different levels of ligand binding (expressed as ligand binding rate, s⁻¹, which is a term proportional to ligand concentration) (Fig. 6B). This quantitative model recapitulates the experimental nanodomain confinement behaviour of EGFR given energy barrier parameters that result in relatively slow EGFR entry and exit from tetraspanin nanodomains (mean entry time 83 s ± 8 s) (Fig. 6B). The proportion of all EGFRs (i.e. both unliganded and ligand-bound) and ligand-bound EGFR that are confined are both largely independent of the EGF binding rate to unliganded EGFR in tetraspanin domains, with the fraction of all EGFR that is confined much lower than the fraction of ligand-bound EGFR that is confined. This model also shows that as the EGF binding rate increases, the fraction of EGFR confined in tetraspanin nanodomains substantially decreases and the fraction of EGFR in clathrin domains substantially increases, which is again consistent with the EGF-stimulated depletion of EGFR from tetraspanin nanodomains and enrichment of EGFR within clathrin nanodomains observed experimentally (Fig. 2). This modeling approach thus supports our observations that EGFR undergoes a shift the molecular mechanism of confinement upon EGF stimulation, without an apparent shift in the total fraction of EGFR that is confined and/or immobile.

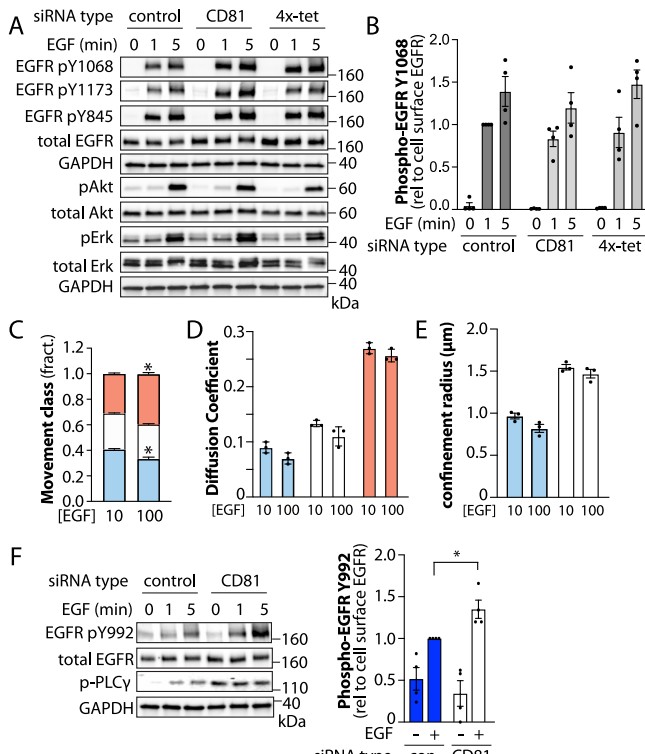

**Fig. 7 | Silencing specific tetraspanins regulates EGFR signaling.** ARPE-19 cells were treated with siRNA to silence CD81 or CD9, CD81, CD82 and CD151 concomitantly (4x-tet), or non-targeting siRNA (control). Following silencing, cells were stimulated with 10 ng/mL EGF as indicated for 1–5 min. **A** Western blots showing various phosphorylated or total protein levels or GAPDH (loading control). **B** Mean ± SE along with measurements from $n = 4$ individual experiments (dots) of phospho-EGFR (Y1068), relative to cell surface EGFR in each condition (Supplementary Fig. 10) are shown. **C–E** ARPE-19 cells were subjected to SPT using 10 ng/mL EGF-Cy3B to label only ligand-bound EGFR in the absence (10 ng/mL EGF total) or presence of an additional 90 ng/mL unlabelled EGF (100 ng/mL EGF total). **C** The mean ± SE of the fraction of EGFR tracks in each mobility category (immobile, confined, mobile) under each condition. Also shown are mean ± SE of diffusion coefficient (**D**) or the confinement radius (**E**). Data is from $n = 3$ independent experiments. Each experiment involved detection and tracking of >500 EGFR objects. $^*p < 0.05$. **F** Western blots of ARPE-19 cells treated with siRNA and stimulated as in (**A**, **B**) probed for phospho-EGFR (Y992), phospho-PLCγ1 or other proteins as indicated (left panels). Also shown are the mean ± SE along with measurements from $n = 4$ individual experiments (dots) of phospho-EGFR (Y992), relative to cell surface EGFR in each condition (Supplementary Fig. 10); EGF. $^*p < 0.05$. For panels showing western blots, approximate molecular weight is shown in kD. Statistical analysis and p-values are indicated in Supplementary Table 1. Source data are provided as a Source Data file.

## Tetraspanins regulate EGFR ectodomain conformation and EGFR signaling

The confinement of unliganded EGFR in tetraspanin nanodomains and the requirement of tetraspanins for ligand binding suggest that tetraspanin nanodomains control EGFR signaling. We examined the effect of perturbation of specific tetraspanin proteins on EGF-stimulated EGFR phosphorylation at low doses of EGF (10 ng/mL). In all cases, we consider the levels of phosphorylated EGFR relative to cell surface EGFR (Supplementary Fig. 10). Silencing of CD81 does not alter EGF-stimulated EGFR phosphorylation at Y1068, Y1173, and Y845, nor phosphorylation of Akt and Erk that depend on EGFR Y1068 and/or Y1173 phosphorylation (Fig. 7A, B). That silencing CD81 elicits a robust reduction of ligand binding (Fig. 4B) yet has a modest effect of EGF-stimulated EGFR phosphorylation suggests that other mechanisms are engaged to propagate signaling in tetraspanin-silenced conditions.

We thus considered the regulation and role of EGFR oligomerization upon EGF stimulation, which contributes to lateral signal propagation of EGFR to unliganded receptors[32–34]. Oligomerization occurs via face-to-face interactions of EGFR dimers, involving the unliganded subunit within a ligand-bound dimer and another unliganded EGFR dimer[34]. Hence, while oligomers form at lower EGF concentrations (~ 10–100 ng/mL) in which only a subset of EGFRs are ligand bound, oligomers are disrupted at higher [EGF] (200–1000 ng/mL) due to depletion of unliganded EGFRs[34]. At 100 ng/mL EGF (10 ng/mL EGF-Cy3B supplemented with 90 ng/mL unlabelled EGF) where 70–90% of ligand-binding sites are occupied (Supplementary Fig. 1F), ligand-bound EGFR is significantly less confined than at 10 ng/mL EGF-Cy3B alone (Fig. 7C), while EGFR diffusion coefficient (Fig. 7D) and confinement radius (Fig. 7E) was indistinguishable. This is consistent with the formation of EGFR oligomers at low [EGF] that together with clathrin nanodomains contribute to confinement of ligand-bound EGFR.

This further suggests that loss of CD81, which substantially reduces EGF binding, may not impact EGF-stimulated EGFR Y1068 phosphorylation because of enhanced contribution of EGFR oligomerization to signaling upon CD81 silencing. To probe this possibility directly, we examined the effect of CD81 silencing on EGFR Y992 phosphorylation. In contrast to other sites on EGFR, the phosphorylation of Y992 occurs with a strong preference in trans between dimers in an oligomer, given the steric constraints of phosphorylation of this kinase domain-proximal residue in cis within a single dimer unit[34]. Silencing of CD81 significantly increased EGF-stimulated Y992 EGFR phosphorylation and also led to enhanced phosphorylation of PLCγ, which occurs subsequent to Y992 activation[61,69] (Fig. 7F). This supports a model in which loss of CD81 reduces EGFR ligand binding affinity, while also increasing the propensity for non-ligand-bound EGFRs to engage in oligomerization with ligand-bound EGFR, leading to lateral signal propagation. This is in turn consistent with our observation of small-order assemblies of EGFR and tetraspanins that may thus function to restrict EGFR oligomerization and lateral signal propagation, in addition to tetraspanins promoting ligand binding, perhaps via larger-scale assemblies that are likely tetraspanin nanodomains.

We next sought to further probe the mechanisms that trigger confinement of ligand-bound EGFR. To do so, we examined the effect of treatment with the EGFR tyrosine kinase inhibitor erlotinib. In addition to inhibition of the kinase activity of EGFR, erlotinib binding to the receptor elicits conformational rearrangement that favors formation of EGFR asymmetric kinase domain dimers which disfavour oligomerization[31] and increase ligand-binding affinity[70]. Consistent with this, cells treated with erlotinib exhibited an decreased binding of the conformationally-sensitive mAb806 (Fig. 8A), as had been reported previously for other type I TKIs that bind the active kinase conformation[14].

As erlotinib alone increases ligand binding affinity[70], and disrupts oligomer formation, we expect that erlotinib treatment will result in higher mobility of ligand-bound EGFR. Consistent with this, erlotinib treatment significantly reduced the confinement of ligand-bound EGFR (Fig. 8B). In contrast, erlotinib was without effect on the confinement of total EGFR, either in the absence or presence of ligand (Fig. 8C–D). Similar effects were observed with other EGFR tyrosine kinase inhibitors (Supplementary Figure 11). Erlotinib also did not impact the diffusion coefficient (Fig. 8E, F) or confinement radius (Fig. 8H, I) of total EGFR, yet increased the diffusion of ligand-bound EGFR in confined and mobile receptor cohorts (Fig. 8G) and increased the confinement radius of confined EGFR (Fig. 8I). Similar results were obtained with erlotinib in other cell lines (Supplementary Fig. 12). The treatment of erlotinib thus leads EGFR to exhibit nearly identical mobility characteristics regardless of tracking via EGF-Cy3B or Fab-Cy3B, both in terms of the fraction of receptors in specific mobility classes, as well as diffusion coefficient and confinement radius of each

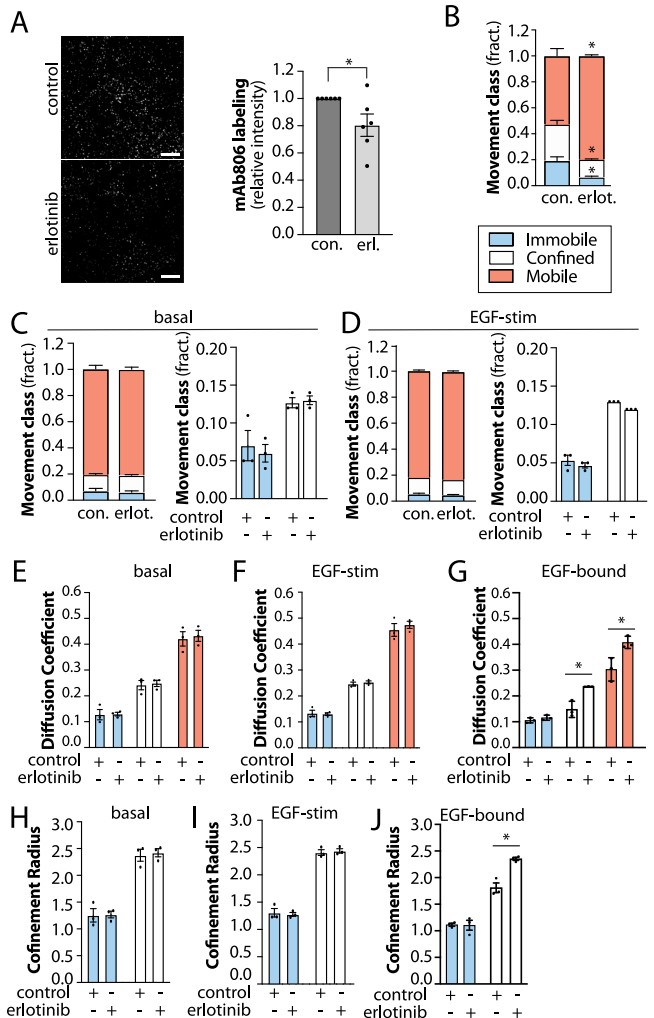

**Fig. 8 | Tyrosine kinase inhibitors selectively alter confinement of ligand-bound EGFR. A** ARPE-19 cells were pre-treated with 2 μM erlotinib for 20 min, followed by stimulation with EGF as indicated for 5 min. Intact (non-permeabilized) cells were incubated with mAb806 at 37 °C for 5 min, followed by fixation and washing of unbound antibodies. Shown are representative images obtained by TIRF microscopy, as well as quantification of mAb806 binding. Scale 20 μm. **B–J** Results of SPT analysis following 20 min 2 μM erlotinib pre-treatment, as indicated. The cells were then subjected to SPT using either Fab-Cy3B to label total EGFR in the absence (**C, E, H**) or presence (**D, F, I**) of unlabelled EGF, or labelled using EGF-Cy3B (**B, G, J**), to label only ligand-bound EGFR. Show in **B–D** are the mean ± SE of the fraction of EGFR tracks in each mobility category (immobile, confined, mobile) under each condition, as well as in **A, B** the same data re-plotted to view the immobile and confined fractions. Also shown are mean ± SE of diffusion coefficient (**E–G**) or the confinement radius (**H–J**). EGF-Cy3B (ligand-bound) data and Fab-Cy3B (total EGFR) data are each from 3 independent experiments, each involving detection and tracking of >500 EGFR objects. *$p < 0.05$. Statistical analysis and p-values are indicated in Supplementary Table 1. Source data are provided as a Source Data file.

mobility class (Fig. 8). This provides yet further support for the conclusion that the differences in mobility of ligand-bound and non-ligand-bound EGFR (e.g. Fig. 1) reflect differences in the behaviour of these receptor classes and not artificial differences resulting from label-based alterations in EGFR mobility. These results also indicate that erlotinib treatment triggers a conformational change throughout the cell surface pool of EGFR that increases the receptor's ligand binding affinity, mimicking that which is typically restricted to EGFR confined within tetraspanin nanodomains in the absence of erlotinib. Following ligand binding in the presence of erlotinib, ligand-bound EGFR remain highly mobile, as erlotinib impairs EGFR oligomerization

but not dimerization[31] and prevented engagement with clathrin-coated pits.

## Discussion

We resolved that only a small fraction of total EGFR has preferential ligand binding at a physiological concentration of EGF, and that ligand binding occurs preferentially, but perhaps not exclusively, in a subset of total EGFR located within tetraspanin nanodomains. The tetraspanin CD81 is required for recruitment of EGFR to confined and immobile fractions in the basal state, while confinement of ligand-bound EGFR was dependent on clathrin and EGFR oligomerization. We also show that loss of CD81 leads to a reduction in ligand binding by EGFR, which modulates EGFR signaling.

EGFR is capable of oligomerization, both in the basal state[31] and upon binding ligand[33]. Oligomers of EGFR could be mobile yet have a reduced diffusion coefficient for smaller oligomers or exhibit outright confinement for sufficiently large oligomers. Our SPT analysis revealed that only a fraction of total EGFR is classified as confined or immobile in the basal state. Importantly, erlotinib treatment results in the formation of dimers that disrupts the auto-inhibitory EGFR oligomers formed solely by unliganded receptors observed in the basal state in some cells[31]. Erlotinib treatment did not alter the confinement of non-ligand bound EGFR (Fig. 8), indicating that in the basal state, the confined and immobile EGFR subpopulations that we observe are likely not the result EGFR homo-oligomerization. We observed similar effects in ARPE-19 non-cancerous cells of epithelial origin, as well as MDA-MB-231 and SUM-149PT breast cancer cells (Supplementary Fig. 12). Hence, EGFR basal oligomers may occur preferentially in specific cell types. Nonetheless, our data show a role for CD81 in the recruitment of EGFR within confined and immobile fractions in the basal state.

Our SPT data shows that ligand-bound EGFR is substantially more confined and/or immobilized than non-ligand-bound EGFR (Fig. 1B). Furthermore, we found that the fraction of ligand-bound EGFR within each mobility class is insensitive to perturbation of tetraspanins (Fig. 3 and Supplementary Fig. 5−9) but instead impacted by perturbation of clathrin (Fig. 5). These results indicate that the molecular basis of confinement of the ligand-bound EGFR is distinct from the non-ligand-bound receptor, and partly dependent on clathrin-coated pits. As formation of EGFR oligomers requires a pool of unliganded EGFR[34], and since ligand-bound EGFR exhibited higher mobility upon treatment with 100 ng/mL EGF (reducing the unliganded EGFR pool) compared to 10 ng/mL EGF (Fig. 7C), this suggests that EGFR oligomers contribute to confinement of ligand-bound EGFR at physiological [EGF]. EGFR homo-oligomers can range from 18.5 nm (tetramer) to 50 nm (12-mer) in length[31], which is smaller than the -100–200 nm diameter of tetraspanin nanodomains and clathrin-coated pits, yet consistent with receptor confinement. In addition, it is difficult to dissect the specific contributions of EGFR oligomerization and recruitment to clathrin-coated pits to confinement of ligand-bound EGFR, as the increased clustering of endocytic adaptor motifs (as in EGFR oligomers) robustly enhances formation of clathrin-coated pits[71]. While we expect that CD81 silencing will lead to larger EGFR homo-oligomers, the mobility of the immobile/confined ligand-bound EGFR may not be appreciably different upon CD81 silencing due to confinement of CD81 oligomers within clathrin structures, with the latter driving mobility characteristics of ligand-bound EGFR. Hence, our results are consistent with the high degree of confinement of ligand-bound EGFR resulting from both EGFR oligomerization and/or recruitment into clathrin-coated pits.

In addition to the role of the tetraspanins CD81, CD82, CD151 and CD9, as well as clathrin and receptor oligomerization in regulating specific aspects of EGFR mobility as we resolve here, there may be other mechanisms that also regulate EGFR mobility at the plasma membrane. For example, other protein-driven membrane structures

such as flotillin nanodomains[72] or association of EGFR with caveolin-1[73], either outside or within caveolae may also regulate EGFR mobility in certain contexts. While we did not observe robust association of EGFR with cell surface flotillin or caveolin-1 structures here (Supplementary Fig. 3F, G), this study is focused on cells that express little or no HER2, which heterodimerizes with EGFR and may alter signaling. Indeed we recently found that HER2 expression renders EGFR signaling dependent on flotillin-1, suggesting that expression of HER2 may impact the mechanisms that regulate EGFR cell surface nanodomain confinement[74]. Hence, while there may be other contexts in which EGFR nanodomain confinement and mobility are regulated, in multiple cells that express EGFR and little HER2 including ARPE-19 and MDA-MB-231 cells, we reveal regulation of EGFR mobility and signaling by tetraspanin nanodomains and clathrin structures.

A common view of tetraspanin nanodomains is that different tetraspanins co-assemble into common structures and that some tetraspanins are functionally redundant (e.g. CD9/81 and CD151)[37–39,75–77], although this is not the case for all tetraspanins as CD53 and CD81 assemble into distinct structures[78]. The tetraspanins CD82, CD81, and CD9 interact with EGFR[39,40] (Supplementary Fig. 2A), and previous reports, based largely on overexpression approaches, suggest that tetraspanins may regulate EGFR signaling[41–46], including regulation of ligand-induced dimerization of EGFR by overexpression of CD82[45]. Other studies showed that silencing CD82 altered EGFR mobility and Erk activation upon stimulation with supra-physiological EGF concentrations[19]. These results are consistent with that of this study in that tetraspanin proteins regulate receptor signaling[37,60], and our work reveals that the regulation of EGFR signaling occurs at the level of ligand binding and a selective role for tetraspanins in confinement of unliganded EGFR. Hence, while previous studies revealed that tetraspanins may also regulate EGFR trafficking, in particular when overexpressed, this work reveals a role for tetraspanin proteins at endogenous expression levels in the regulation of EGFR ligand binding.

Our experiments also reveal that in addition to the outright confinement of EGFR in the basal state that was dependent on CD81 (Fig. 3), silencing of all tetraspanins examined also increased the diffusion coefficient of mobile EGFR in the basal (unliganded) or ligand-bound states (Fig. 3, Supplementary Fig. 5–9). This suggests that EGFR also associates with some tetraspanins in smaller protein assemblies that nonetheless retain mobile behaviour. Consistent with this, we also observe a range of CD81 structure sizes based on intensity of TIRF (Supplementary Fig. 3I) and STED (Supplementary Fig. 3K) microscopy, as well as significant EGFR association with CD81 objects over a range of sizes. We also observe that EGFR association with CD81 is retained following EGF stimulation (Fig. 2B), which is consistent with the co-enrichment of EGFR and CD81 within clathrin-coated pits following EGF stimulation (Fig. 5J, K). One possible explanation for this observation is that smaller-order EGFR-CD81 assemblies are retained upon ligand binding or that clathrin-coated pits may form at predetermined specific sites of CD81-positive tetraspanin nanodomains, such that CD81 becomes dispensable for confinement of ligand-bound EGFR; this will require additional study. Nonetheless, our results show that CD81 is selectively required for the confinement of a small cohort of unliganded EGFR.

Our results show that tetraspanins regulate EGFR confinement and signaling, sequestering a minor subpopulation of EGFR alongside factors that promote high affinity ligand-binding with tetraspanin nanodomains. This model is consistent with many previous studies that show that the ligand-binding of EGFR is regulated, and that EGFR exhibits subpopulation of receptors at the cell surface with distinct affinities[4–10].

Silencing CD81 and other tetraspanins resulted in loss of ligand binding (relative to cell surface EGFR) (Fig. 4B) yet was without effect on EGF-stimulated EGFR autophosphorylation of Y1068 and Y1173

(Fig. 7A, B). The ability of a ligand-bound active EGFR dimers to oligomerize with unliganded EGFR dimers extends the phosphorylation of EGFR laterally, with phosphorylation of the C-terminal tails occurring in trans between dimer units in an oligomer[34]. That the substantial loss of ligand binding by EGFR upon silencing of CD81 and other tetraspanins does not impact EGFR phosphorylation on Y1068 indicates that the reduction of ligand-bound EGFRs in CD81-silenced cells is offset by enhanced oligomerization-dependent signaling compared to that present in unperturbed cells. Consistent with this, silencing of CD81 increased EGF-stimulated Y992 phosphorylation that is supported largely by phosphorylation in trans within an EGFR oligomer, and not in cis within a single EGFR dimer[34].

Collectively, this suggests that CD81 and possibly other tetraspanins regulate two aspects of EGFR signaling: (i) enhancement of direct EGFR binding to EGF, likely restricted to the confined fraction of EGFR in larger-scale tetraspanin assemblies and (ii) suppression of formation of signaling oligomers of EGFR, perhaps because of small-scale EGFR-tetraspanin assemblies that are more mobile. This reveals that tetraspanins have a key role in control of receptor signaling, both in terms of its magnitude and sensitivity, but also in modulation of relative phosphorylation of specific EGFR tyrosine residues.

Following EGF binding, a subset of which occurs preferentially in tetraspanin nanodomains, EGFR undergoes recruitment to clathrin structures at the cell surface, leading to eventual internalization of EGFR. While clathrin perturbation did not impair EGF-stimulated EGFR phosphorylation, clathrin structures are enriched in downstream signaling proteins and modulate Akt signaling by EGFR[52,53,62,79] suggesting that clathrin-dependent regulation of Akt occurrs subsequent to EGFR phosphorylation, and thus subsequent to tetraspanin-dependent regulation of EGFR ligand binding.

While the regulation of the transition from a tethered, low-affinity conformation of the ectodomain to an untethered conformation has been suggested from molecular dynamic simulations and study of EGFR mutants[14,65,80], the cellular processes that contribute to the regulation of these conformational transitions were poorly understood. Here, we reveal that the ectodomain conformation and/or protein-protein interactions of EGFR may be gated by specific tetraspanins in a manner consistent with transition from a tethered, low-affinity EGFR to a higher affinity, untethered EGFR ectodomain in a spatially-restricted manner within tetraspanin nanodomains. This work reveals a mechanism for the regulation of EGFR conformation and ligand binding within specific subpopulations of EGFR, which has important impact on the understanding EGFR regulation of cell physiology, as well as the role of EGFR as a driver of tumor progression.

## Methods

### Materials

Phorbol 12-myristate 13-acetate (PMA) was obtained from Abcam (Cambridge, MA). Erlotinib, gefitinib, and lapatinib were obtained from MedChemExpress (Monmouth Junction, NJ). mAb806 isolated from hybridoma supernatant[14,65–67]. Primary antibodies used for nanodomain immunofluorescence labeling experiments are described in Supplementary Table 2; CD81, CD82 and CD151 antibodies were validated by siRNA silencing (Supplementary Figure 2). Fluorophore-conjugated or secondary antibodies are described in Supplementary Table 3. For immunoblotting, primary and horseradish peroxidase (HRP)-conjugated secondary antibodies are described in Supplementary Tables 2 and 3, respectively. Other reagents used in this study are listed in Supplementary Table 4.

### Cell culture and stable cell line generation

Wild-type retinal pigment epithelial cells (ARPE-19, male; RPE cells herein), and ARPE-19 cells stably expressing clathrin light chain fused to enhanced GFP (RPE-GFP-CLC)[61,68,81] were cultured in DMEM/F12 (ThermoFisher) supplemented with 10% fetal bovine serum (FBS,

ThermoFisher), 100 U/ml penicillin, and 100 µg/ml streptomycin (P/S, Life Technologies) at 37 °C and 5% CO2. MDA-MB-231 cells (female) were maintained in DMEM high glucose (ThermoFisher) supplemented with 10% FBS and P/S. SUM-149PT cells (female) were maintained in Ham's F-12 media supplemented with 10% FBS, P/S, 1 µg/ml hydrocortisone, 5 µg/ml human insulin, and 10 mM HEPES. Cell lines were initially obtained from American Tissue Type Collection (ATCC, Manassas, VA).

To generate stable cells with inducible expression of SNAP-tagged EGFR, we used the sleeping-beauty transposon system[59]. pSBtet-BP was a gift from Eric Kowarz (Addgene plasmid # 60496; http://n2t.net/addgene:60496; RRID:Addgene_60496)[59]. pCMV(CAT)T7-SB100 was a gift from Zsuzsanna Izsvak (Addgene plasmid # 34879; http://n2t.net/addgene:34879; RRID:Addgene_34879)[82]. An oligonucleotide encoding SNAP fused to EGFR were generated by BioBasic Inc (Markham, ON, Canada) and inserted into pSBtet-BP, by fusing the following sequences: (i) the first 93 nucleotides of the EGFR ORF (as per NM_005228) that encode the first 31 amino acids (including the signal sequence encoded in the first 24 amino acids), followed by (ii) the ORF sequence of the SNAP tag, followed by (iii) a linker/spacer peptide (GGA GGT AGT GGA GGT GCA AGT GCT), followed by (iv) the rest of the ORF sequence of human EGFR (starting at the nucleotide sequence encoding amino acid residue 25, or the nucleotide sequence CTG GAG GAA GGA AAA GTT TGC CAA) that immediately follows the signal sequence (amino acids 1–24).

ARPE-19 cells stably expressing N-SNAP-EGFR under control of the Sleeping Beauty Transposon system[59] were generated[79] and were selected for 2 weeks in 2 µg/ml puromycin. N-SNAP-EGFR expression was induced by treating cells with 500 nM doxycycline in complete DMEM/F12 for 96 h prior to experiments. Supplementary Fig. 1F shows that the level of induction of N-SNAP-EGFR under these conditions was ~2-3 fold that of endogenous, thus limiting potential artifacts of EGFR overexpression.

### Single particle labeling and tracking of EGFR

The Fab fragment was generated from mAb108[55]. The mAb108 hydridoma was obtained from American Type Culture Collection (ATCC, catalog. HB-9764, Manassas, VA). Generation of Fab fragments and Fab labeling with Cy3B was performed by AbLab Biologics, to produce Fab-Cy3B (University of British Columbia, BC, Canada). EGF-Cy3B was generated using NHS-Cy3B (Cytiva, Marlborough, MA) conjugated to recombinant human EGF (Gibco, ThermoFisher Scientific), using 1:1 labeling stoichiometry[63]. To label single particles (unless otherwise indicated), cells were incubated with either 10 ng/mL EGF-Cy3B for 5 min prior to imaging (to selectively label and track ligand-bound EGFR), or with 50 ng/mL Fab-Cy3B (to label and track total EGFR) for 10 min followed by washing to remove unbound antibodies, in media lacking EGF or following 5 min of stimulation with 10 ng/mL unlabelled recombinant EGF. To label N-SNAP-tagged EGFR in ARPE-19 cells, cells were incubated with 5 nM SNAP-Surface-488 (New England BioLabs Inc., Ipswich, MA) and either Fab-Cy3B in reduced serum DMEM/F12 for 30 min followed by 3x media wash or EGF-Cy3B as described above. Time-lapse imaging was performed using total internal reflection fluorescence (TIRF) microscopy using a Quorum (Guelph, ON, Canada) Diskovery instrument, comprised of a Leica DMi8 microscope equipped with a 63×/1.49 NA TIRF objective with a 1.8× camera relay (total magnification 108×). Imaging was done using 561 nm laser illumination and 620/60 nm emission filters and acquired using an iXon Ultra 897 EM-CCD (Spectral Applied Research, Andor, Toronto, ON, Canada) camera at a framerate of 20 Hz, for total length of time-lapse 250 frames. All live cell imaging was performed using cells incubated in serum-free DMEM/F-12 or DMEM without phenol red or P/S. Single particles labelled with Cy3B were detected and tracked[56], and the motion types, diffusion coefficients and confinement radii were determined using moment-scaling spectrum analysis (MSS)[57,83]. The

term mobile as used in this work refers to the "free diffusion" motion type as classified by MSS. Of note, in MSS, motion types are determined by how the particle movement scales with time (lag), regardless of the value of the diffusion coefficient or confinement radius.

### EGFR nanodomain microscopy and analysis

After treatments as indicated, cells were fixed with 4% paraformaldehyde for 30 min, followed by quenching of fixative in 100 mM glycine, cell permeabilization in 0.1% Triton X-100 (all solutions made in PBS), and then blocking in Superblock Blocking Buffer (Thermo Fisher Scientific). Subsequently, cells were stained with primary and secondary antibodies as indicated, and then subjected to imaging using TIRF microscopy using a Quorum (Guelph, ON, Canada) Diskovery instrument, comprised of a Leica DMi8 microscope equipped with a 63×/1.49 NA TIRF objective with a 1.8× camera relay (total magnification 108×). Imaging was done using 405-, 488-, 561-, and 637-nm laser illumination and 450/55, 525/50, 620/60, and 700/75 emission filters and acquired using a Zyla 4.2Plus sCMOS camera (Hamamatsu). Image acquisition was performed in MetaMorph 7.10.3.279 (Molecular Devices, San Jose, CA).

Systematic, unbiased detection and analysis of diffraction-limited EGFR objects in fixed cells (e.g. Fig. 2), was performed using custom software developed in Matlab (Mathworks Corporation, Natick, MA)[52,61,63,68].

Briefly, for analyses involving intensity (Figs. 2B–E, 4A, 5J, S3), diffraction-limited EGFR objects (resulting from labelling by Fab-Cy3B or EGF-Cy3B) were detected using a Gaussian-based model method to approximate the point-spread function of EGFR ('primary' channel). The TIRF intensity corresponding to various proteins in a 'secondary' (or 'tertiary') channel (eGFP-clathrin, CD82, caveolin, or flotillin) within EGFR objects was determined by the amplitude of the Gaussian model for the appropriate fluorescence channel for each object detected in the 'primary' channel. As such, the measurements of fluorescently labelled proteins within EGFR objects represent their enrichment relative to the local background fluorescence in the immediate vicinity of the detected EGFR objects. For analysis of CD81 intensity as per Supplementary Fig. 3I, J, the frequency distribution of CD81 intensity within CD81 detected objects was determined within the 5th to 95th percentile of CD81 intensities in each experiment, then binning into 26 cohorts. To sort CD81 objects into CD81-low and CD81-high, the median CD81 object intensity was used as an arbitrary but systematic threshold.

For colocalization analyses not involving intensity (Fig. 2F–H and Fig. 5K), diffraction-limited objects in all channels (EGFR and others) were detected as described above for EGFR. Object colocalization was then assessed based on the nearest neighbor distances between the detected object locations, as described in ref. 64, using a colocalization radius of 3 pixels. For conditional colocalization analysis (Fig. 5K), colocalization between CD81 and clathrin was assessed similarly, but after subdividing the CD81 and clathrin objects based on their colocalization with EGFR, as described in more detail in ref. 58.

### Stimulated emission depletion microscopy

After treatments as indicated, wild-type ARPE-19 cells were washed with PBS, then fixed in 4% paraformaldehyde in PBS for 10–15 min at room temperature. Following fixation, cells were washed, then permeabilized with PBS supplemented with 0.1% Tween-20 (PBST). Cells were blocked in 10% fetal bovine serum in PBST (blocking solution) for 1 h at room-temperature, followed by primary antibody incubation for CD81 (diluted 1:100) in blocking solution overnight at 4 °C. The next day, cells were incubated with Alexa Fluor 488-conjugated secondary antibody (diluted 1:100) in 1% serum in PBST for 1 h at room temperature. After final washes, coverslips were mounted onto glass slides using ProLong Diamond Antifade Mountant (Invitrogen).

Stimulated emission depletion (STED) microscopy was performed using a Leica TCS SP8 STED 3X microscope using HyD detectors and a 100x/1.4 NA oil objective. Samples labelled with Alexa Fluor 488- and Cy3-conjugated secondary antibodies (see Supplementary Table 3) were excited using a white light laser at 499 nm. Emissions were time-gated 0.50–6.00 (green) or 1.00–6.00 ns (red). 1.5 W depletion lasers at 592 nm were used at 21% or 70% maximal power for green and red channels, respectively. Images were acquired and deconvolved using Leica LAS X software with Lightning Deconvolution.

### siRNA gene silencing
siRNA transfections were performed using custom-synthesized siRNAs using RNAiMAX transfection reagent (Life Technologies) as per manufacturer's instructions. Briefly, each siRNA was transfected at a concentration of 220 pmol/mL with transfection reagent in Opti-MEM Medium (Life Technologies). Cells were incubated with the siRNA complexes for 4 h, after which cells were washed and replaced in regular growth medium. siRNA transfections were performed twice (72 and 48 h) before each experiment. Sequences used were as follows (sense): control (non-targeting): CGU ACU GCU UGC GAU ACG GUU; clathrin heavy chain (clathrin): GGG AAU AGU UUC AAU GUU U; CD82: CCC AUC CUG ACU GAA AGU AUU; CD9: CCA CAA GGA UGA GGU GAU UUU; CD81: CCU CAG UGC UCA AGA ACA AUU; CD151: CCC AUC CUG ACU GAA AGU AUU.

### Immunoblotting
After transfection, treatment with inhibitors, and/or stimulation with EGF, whole-cell lysates were prepared using in 5X Laemmli sample buffer (LSB; 0.5 M Tris, pH 6.8, glycerol, 10% SDS, 10% β-mercaptoethanol, and 5% bromophenol blue; all from BioShop, Burlington, Canada) supplemented with a protease and phosphatase inhibitor cocktail (1 mM sodium orthovanadate, 10 nM okadaic acid, and 20 nM Protease Inhibitor Cocktail, each obtained from BioShop). Lysates were then heated at 65 °C for 15 min and passed through a 27.5-gauge syringe. Proteins were resolved by glycine-Tris SDS–PAGE followed by transfer onto a polyvinylidene fluoride membrane[78]. Western blot signals to detect the intensity corresponding to phosphorylated proteins (e.g., pEGFR) were obtained by signal integration in an area corresponding to the appropriate lane and band for each condition. This measurement is then normalized to the loading control (e.g., actin) signal[61].

### Quantitative modeling
EGFR diffuses on a square, two-dimensional lattice with diffusivity $D = 0.2 \, \mu m^2/s$[84]. The square region is 0.2 μm x 0.2 μm with periodic boundary conditions. The lattice has sites separated by $\Delta x = 0.002 \, \mu m$, and the timestep used is $\Delta t = 5 \times 10^{-6} \, s$, determined by the two-dimensional requirement that $\Delta t = (\Delta x)^2/(4D)$[85]. Each timestep, the EGFR attempts to step to one of the four nearest-neighbor lattice sites. If there is no energy barrier, the step is successful; if there is an energy barrier between the two lattice sites, the step is successful with a probability determined by the Metropolis criterion of $P_{success} = \exp[-\Delta E/(k_B T)]$, where $\Delta E$ is the energy barrier height[86].

The square membrane has 10 circular tetraspanin domains and 10 circular clathrin domains, each of which do not overlap other tetraspanin or clathrin domains. At the beginning of each EGFR trajectory these 20 domains are randomly positioned. Each domain has a radius of 0.0056 μm, so that each domain is approximately 100 nm². The energy barrier for an unliganded EGFR to enter a tetraspanin domain is $\Delta E_{et} = 11.5 \, k_B T$, for an unliganded to leave a tetraspanin $\Delta E_{lt} = 13 \, k_B T$, for a liganded EGFR to enter a clathrin domain $\Delta E_{ec} = 1 \, k_B T$, and for a liganded EGFR to leave a clathrin domain $\Delta E_{lc} = 4.5 \, k_B T$. These energy barriers have been selected to provide EGFR confinement consistent with experimental measurements. Energy barriers are not encountered for transitions of liganded EGFR into and out of tetraspanin domains and of unliganded EGFR into and out of clathrin domains.

Unliganded EGFR in a tetraspanin domain become ligand-bound at a rate that is varied (the 'EGF binding rate' in Fig. 6B). Liganded EGFRs in a clathrin domain are removed from the membrane at a rate of $0.05 \, s^{-1}$. EGFR begin each simulation without ligand and outside tetraspanin and clathrin domains. The fraction of time until EGFR removal that each individual EGFR trajectory is liganded and unliganded within tetraspanin and clathrin domains is tracked. The fraction of time after ligand binding that each individual EGFR trajectory is within a clathrin domain is also tracked. The average EGFR confinement values, overall and ligand-bound, are determined by weighting the fractions by the time until EGFR removal from the membrane, as the likelihood of experimental observation of an EGFR is proportional to its lifetime on the membrane.

### Statistical analysis
All measurements were subjected to statistical testing as described in Supplementary Table 1 with a threshold of $p < 0.05$ for statistically significant differences between conditions. All statistical tests were performed using GraphPad Prism 9.0 (GraphPad, Boston, MA) except for analysis shown in Fig. 2F–H and Fig. 5K, which were performed using MATLAB (MathWorks, Natick, MA).

### Reporting summary
Further information on research design is available in the Nature Portfolio Reporting Summary linked to this article.

## Data availability
The datasets generated during and/or analysed during the current study are available from the corresponding author on request. The amount of data represented by microscopy image files exceeds our capacity for providing access to this data in a public repository. Source data are provided with this paper.

## Code availability
The analysis of single particle tracking was done in Matlab using uTrack software[56] available here: https://github.com/DanuserLab/u-track. The intensity-based co-localization analysis within nanodomains or EGFR objects detected in TIRF microscopy images was done in Matlab using the runDetection function, part of the cmeAnalysis pipeline[68], available here: https://github.com/DanuserLab/cmeAnalysis. Additional applications and validation of this method were as described in[79]. The colocalization analysis not based on intensity was done in Matlab using 3-color conditional colocalization analysis algorithm[58], available here: https://github.com/kjaqaman/conditionalColoc.

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

## Acknowledgements

Funding for this research was provided by a Project Grant (PJT-156355) from the Canadian Institutes of Health Research (CIHR) to C.N.A. and G.D.F., and a New Investigator Award from the Canadian Institutes of Health Research (CIHR), Funding from Toronto Metropolitan University, as well as an Early Researcher Award (Ontario Ministry of Research, Innovation and Science) to C.N.A., A.I.B. was supported by a Natural Sciences and Engineering Research Council of Canada Discovery Grant 2021-03431 and start-up funds provided by the Toronto Metropolitan University Faculty of Science. A.M.S. was supported by a National Health and Medical Research Council (NHMRC, Australia) Investigator grant (No. 1177837). G.D.F. holds a Canadian Research Chair in Multiomics of Lipids and Innate Immunity. We thank Austin R. Miranda for providing technical assistance with MATLAB workflow optimization.

## Author contributions

Conceptualization: M.G.S., G.D.F, C.N.A. Methodology: M.G.S., A.I.B., J.V.-L., J.P.B., A.M.S., K.J., G.D.F., C.N.A. Software: A.I.B., J. V.-L., K.J. Formal Analysis: M.G.S., A.I.B., K.J, C.N.A. Investigation: M.G.S, A.I.B., J. V.-L. J.P.B., Writing - Original Draft: M.G.S., A.I.B., K.J., C.N.A. Writing - Review and Editing: M.G.S., A.I.B., J.V.-L., J.P.B., A.M.S., K.J., G.D.F., C.N.A. Supervision: K.J., G.D.F., C.N.A. Project Administration: M.G.S., C.N.A. Funding acquisition: K.J., G.D.F., C.N.A.

## Competing interests

The authors declare no competing interests
