## [Peer Review File · Nature Communications]

Confinement of unliganded EGFR by tetraspanin nanodomains gates EGFR ligand binding and signalingREVIEWER COMMENTS

Reviewer #1 (Remarks to the Author):

The manuscript uses single particle tracking (SPT) methods to analyze the mobility of EGFR receptors present at the cell membrane of human retinal pigment epithelial cells. The effects of EGF ligand binding as well as of knocking down clathrin and tetraspanin proteins on the mobility of ligand bound and total EGFR were studied. EGFR receptor mobility and clustering regulates its functions and therefore the questions addressed in the manuscript are of significant relevance. However, the manuscript currently has fundamental problems in experimental design, data presentation and interpretation.

1. Throughout the manuscript, EGF fused to the fluorophore Cy3B (EGF-Cy3B) is used to image the fraction of EGFR that is ligand-bound whereas a Fab fragment that recognizes EGFR conjugated to a fluorophore (Fab-Cy3B) is used to image the total receptor present (irrespective of whether it is EGF-bound or not). In Figure 1B, the fraction of EGF-Cy3B that corresponds to immobile or confined receptors does not significantly change with the concentration of EGF and is about 50%. In Figures 1B and S1E, the fraction of receptor that is immobile or confined does not change either with EGF treatment, even with EGF concentrations of up to 200 ng/ml, and is about 20%. However, the fraction of EGFR that is ligand bound changes drastically in the range of concentrations studied and even reaches 80-90% at the highest concentration of 200 ng/ml (Figure S1F). Why is this not reflected in the fractions of immobile and confined receptor, measured by Fab-Cy3B?

A simple control experiment would be to complement the results in Figure S1E on Fab-Cy3B with measurements of EGF-Cy3B, at the same concentration range. The authors are urged to do this experiment. When almost all the receptor is ligand bound (which is the case at high concentrations of EGF), then the results of mobility obtained from EGF-Cy3B and Fab-Cy3B images need to be similar. If they are not, then a possible explanation is that the EGF-Cy3B molecule affects EGFR mobility in a manner that is different from EGF, which would invalidate the approach. Also, it is necessary to show that treating the cells with EGF-Cy3B in the range of 5-25 ng/ml (which is the range for most experiments) and analyzing with the Fab labeled with a different fluorophore (for example Fab-Cy2) gives the same results as treating the cells with equivalent concentrations of unlabeled EGF.

2. In Figure 1C, the diffusion coefficient of the mobile fraction is stated to significantly increase with EGF treatment (from about 0.4 to above 0.5, as estimated from the plot). The authors say that this suggests that EGF stimulation releases some receptors from small scale interactions that restrict diffusion. However, this increase of diffusion coefficient of the mobile fraction upon treatment with EGF, as measured by Fab-Cy3B, is not detected in Figure 3 (3D vs 3E, control siRNA) or Figure 5 (5D vs 5E, control siRNA). How do the authors reconcile these differences?

3. In Figure 1, scale should be “3 μm ” not “3 μM ”.

4. In Figure 2, is the CD81 signal constant during the 15 min of EGF treatment? (It appears to decrease with time in the images shown in Figure 2A).

5. In Figure 2, is there ligand induced receptor endocytosis during the time scale of these experiments? In other words, is the total number of receptors at the membrane constant? The results in Figure 2 are interpreted as suggesting that there are 2 scales of association (line 206), of EGFR and tetraspanins, which is an over-interpretation of the results.

6. In Figure 3, the same data on “movement class” are presented twice in Figures 3A and 3B (replotted), which is unnecessary.

7. Is it possible that the effects observed upon treatment with of clathrin siRNA are partly due to inhibition of endocytosis?

Reviewer #2 (Remarks to the Author):

Sugiyama et al. report results on using single-particle tracking (SPT) to study spatial regulation of EGFR by a set of tetraspanins, namely CD81, CD82, CD9, and CD151. By labeling ‘total’ EGFR with an anti-EGFR Fab and ‘ligand-bound’ EGFR with EGF, both of conjugated to Cy3B, they studied the diffusion properties of both EGFR populations under both basal conditions and various perturbations. They found that, whereas EGFR (total) is mostly (~80%) mobile, ligand-bound EGFR is much more likely (~50%) in the confined or immobile states (Figure 1). Upon ligand (EGF) stimulation, the amount of EGFR associated the three tetraspanins remained unchanged, so did the amount of CD81 associated with EGFR, but the amount of CD82 and CD151 associated with EGFR decreased significantly (Figure 2). CD81 knock down with siRNA led to a higher fraction of total EGFR in the mobile state (and concomitantly lower fractions in the confined or immobile states) – with or without EGF stimulation, where the mobile EGFR molecules also exhibited faster diffusion, but it had little effect on the diffusion properties of ligand-bound EGFR (Figure 3). Knocking down the tetraspanins (particularly CD81) decreased the fraction EGFR capable of ligand-binding but increased the fraction that binds mAb806, an EGFR antibody with ectodomain conformation selectivity (Figure 4). Interestingly, tetraspanin knockdown had little effect on common tyrosine phosphorylations on EGFR (with the exception of pY992) or on ERK or AKT activation (Figure 7). Blockade of tyrosine phosphorylation of EGFR using TKIs such as erlotinib increased the mobile population of ligand-bound EGFR as well as the mobility of this population but had little effect on the diffusion of total EGFR (Figure 8). Combining these results and modeling (Figure 6), other evidence not

thoroughly summarized here (Figure 5 on clathrin), the authors suggest a model where EGF preferentially binds to EGFR pre-confined in tetraspanin nanodomains and, upon EGF stimulation, leaves these nanodomains but quickly become associated with clathrin-based nanodomains.

Regulation of EGFR by tetraspanins has been studied extensively but, as the authors rightfully put, gaps remain in our understanding of the exact roles of tetraspanins in the complex process of EGFR ligand binding, activation, and signal propagation. Overall, the study is helpful in addressing these gaps, and the use of SPT and subsequent SPT data analysis are well appreciated. The problem with this work, however, is that the authors try to describe the entire process – from baseline EGFR diffusion and confinement to EGF binding selectivity to EGFR translocation (presumably from one nanodomain to another) to signal propagation in a single study essentially with SPT alone. This is a very ambitious undertaking. I am afraid that, even with the large amount of data presented and the great length that the authors have taken to try to reconcile all the observations, most conclusions are ambiguous rather than definitive. After reading the manuscript many times I still could not put all the pieces together. In my opinion, it would have been much better if the authors focused on a single point (such as the role of tetraspanins in ligand gating) and bring it to a close. Below are some of the issues that I had.

First, on the imaging of total vs ligand-bound EGFR. The two populations were probed using Cy3B labeled Fab and EGF, respectively, it was not clear to me whether EGFR populations detected by the two probes overlap or exclude each other. Cy3B-Fab was stated to probe ‘total EGFR’ so my initial assumption was that EGFR detected by Cy3B-EGF should be a subset of that detected by Cy3B-Fab. However, as I read on I became less and less certain about this. For example, in Figure 1C and D, there was not any indication that a subpopulation of the ‘total’ EGFR corresponds to the ligand-bound EGFR (i.e., exhibiting similar diffusion properties). Same thing for Figure 3 E&F, and H&I. While this could be because the ligand-bound population is small, the authors estimated that 10-40% of total EGFR should be ligand-bound at 5-10 ng/mL EGF and 70-90% at 100 ng/mL. At these binding fractions, the diffusion properties (and the changes thereof) of the ligand-bound population should have been more clearly reflected in those of the total EGFR.

Second, on the preferential binding of EGF to EGFR pre-confined in the tetraspanin nanodomains. The main arguments for this conclusion include: (a) EGF stimulation led to decreased amount of tetraspanins associated with EGFR (but not the amount of EGFR associated with tetraspanin); (b) knock down of CD81 decreased the confined and immobile fractions of total EGFR regardless of EGF stimulation, but it did not change the mobility of EGF-bound EGFR; (c) the fraction of ligand-bound EGFR (relative to all probed by Cy3B-EGF) associated with CD82-positive membrane regions was greater than that of the total EGFR (probed by Cy3B-Fab) at time points as early as 15s after incubation with EGF (Cy3B-EGF); and (d) knocking down CD81 or all 4 tetraspanins reduced EGF binding to cell surface. These observations support, but do not prove, that EGF binding preferentially take place in large tetraspanin nanodomains (where a subset of EGFR is confined). Despite (c), the fractions of total or ligand-bound EGFR found on CD82-positive spots were only 10-15% at time points as early as 15s. According to the simulation model presented in Figure 6, the half time for EGFR to exit these tetraspanin domains are 83+/-8s. Does that

mean that the majority of ligand-binding of EGFR (>80%) took place in CD82-negative regions? Additionally, whereas the reduced EGF binding capacity of EGFR after tetraspanin knockdown does suggest a role of tetraspanins in regulating ligand binding (which had been well established), there was no other direct evidence presented to directly prove that the loss of EGF binding was caused by reduced confinement in tetraspanin domains. In Figure 4B, ligand binding was lost by (only) 30-40% when all 4x tetraspanins were knocked down; would this further suggest that tetraspanin domains do not necessarily 'dominate' ligand binding of EGFR?

Third, on the explanation for the lack of effect on EGFR phosphorylation (except pY992) or signaling upon tetraspanin silencing despite significant reduction in ligand binding. The authors suggested that this is likely due to compensation through increased EGFR oligomerization, which 'amplifies' the effect of low ligand binding by cross phosphorylation of receptors in the same oligomer. Again, while this is plausible, there was not enough evidence to prove that this is indeed the case. How EGFR oligomerization compensates the loss in dimerization and subsequent intra-dimer phosphorylation to yield nearly identical receptor phosphorylation and ERK/AKT signaling is unclear. Besides, tetraspanin silencing reduced the confinement of total EGFR (Figure 2) - should that have reduced oligomerization instead?

Fourth, regarding the modeling, which took place before the discussions on oligomers. As a result, the modeling did not take into account receptor oligomerization, making the value of this effort somewhat questionable. While modeling can be very useful in this work, it does require a relatively clear understanding of the critical processes, which is not yet the case. Among other things, tetraspanin domains may not be the only membrane structures regulating diffusion and ligand-binding of unliganded EGFR (this should be taken into account in data interpretation throughout the manuscript, not just the modeling).

Lastly, there were also some inconsistencies in the mobility analyses. In Figure 1, 'total' EGFR diffusing at $\sim 0.5 \mu\text{m}^2/\text{s}$, $\sim 0.2 \mu\text{m}^2/\text{s}$, and $\sim 0.1 \mu\text{m}^2/\text{s}$ was assigned 'mobile', 'confined', and 'immobile', respectively; by contrast, ligand-bound EGFR diffusing at $\sim 0.3 \mu\text{m}^2/\text{s}$, $\sim 0.13 \mu\text{m}^2/\text{s}$, and $\sim 0.11 \mu\text{m}^2/\text{s}$ (read from the plot directly so these numbers are approximate) was assigned the same three states. The 'confined' state of total EGFR has a diffusion rate almost as fast as the 'mobile' state of ligand-bound EGFR. What was the criterion for assigning these states as mobile, confined, or immobile? One would also need to be careful to call a state 'confined' simply based on a small diffusion coefficient; it could be slow but free instead of confined diffusion. Moreover, the tetraspanin domains were supposedly 100-200 nm, yet the 'confinement' radii for all the confined or even the immobile states were on the order 1-3 μm . Could the authors explain the discrepancies?

Sugiyama et al. - Response to reviewer comments

We provide our responses to the reviewer comments in blue text under each specific comment.

Reviewer #1 (Remarks to the Author):

The manuscript uses single particle tracking (SPT) methods to analyze the mobility of EGFR receptors present at the cell membrane of human retinal pigment epithelial cells. The effects of EGF ligand binding as well as of knocking down clathrin and tetraspanin proteins on the mobility of ligand bound and total EGFR were studied. EGFR receptor mobility and clustering regulates its functions and therefore the questions addressed in the manuscript are of significant relevance. However, the manuscript currently has fundamental problems in experimental design, data presentation and interpretation.

We thank the Reviewer for their detailed review of our manuscript and for the insightful comments and suggested revisions. Below, we have addressed each comment by providing additional experiments and/or modifications to the text. The changes we made in response to the comments from the reviewer have allows us to provide supporting lines of evidence and validation of the experimental design and/or further clarity regarding the interpretations presented in the original submission.

1. Throughout the manuscript, EGF fused to the fluorophore Cy3B (EGF-Cy3B) is used to image the fraction of EGFR that is ligand-bound whereas a Fab fragment that recognizes EGFR conjugated to a fluorophore (Fab-Cy3B) is used to image the total receptor present (irrespective of whether it is EGF-bound or not). In Figure 1B, the fraction of EGF-Cy3B that corresponds to immobile or confined receptors does not significantly change with the concentration of EGF and is about 50%. In Figures 1B and S1E, the fraction of receptor that is immobile or confined does not change either with EGF treatment, even with EGF concentrations of up to 200 ng/ml, and is about 20%. However, the fraction of EGFR is that is ligand bound changes drastically in the range of concentrations studied and even reaches 80-90% at the highest concentration of 200 ng/ml (Figure S1F). Why is this not reflected in the fractions of immobile and confined receptor, measured by Fab-Cy3B?

A simple control experiment would be to complement the results in Figure S1E on Fab-Cy3B with measurements of EGF-Cy3B, at the same concentration range. The authors are urged to do this experiment. When almost all the receptor is ligand bound (which is the case at high concentrations of EGF), then the results of mobility obtained from EGF-Cy3B and Fab-Cy3B images need to be similar. If they are not, then a possible explanation is that the EGF-Cy3B molecule affects EGFR mobility in a manner that is different from EGF, which would invalidate the approach. Also, it is necessary to show that treating the cells with EGF-Cy3B in the range of 5-25 ng/ml (which is the range for most experiments) and analyzing with the Fab labeled with a different fluorophore (for example Fab-Cy2) gives the same results as treating the cells with equivalent concentrations of unlabeled EGF.

We thank the reviewer for the opportunity to elaborate on this important point. In the manuscript, we had directly compared the mobility behaviour of EGFR as detected by EGF-Cy3B and Fab-Cy3B at a lower concentrations of EGF (5-25 ng/mL). Under these conditions, we see a clear difference in the confinement of non-ligand-bound EGFR (detected via Fab-Cy3B in the absence of unlabelled EGF) and ligand-bound EGFR (detected via EGF-Cy3B). Together with many additional lines of evidence, this led us to propose the model in which ligand-bound EGFR is a subset of total EGFR, such that the largely confined population of ligand-bound EGFR represents a subset (10-30%) of the total EGFR subpopulation, as detected by Fab-EGFR in the presence of unlabelled ligand.

The reviewer raises a good point that stimulation of EGFR labelled with Fab-Cy3B with unlabelled EGF does not robustly increase the fraction of mobile EGFR (**Fig S1E**). However, as we increase the dose of unlabelled EGF, there is in fact a significant decrease in the fraction of mobile EGFR at 100 ng/mL and 200 ng/mL EGF (compared to basal, no EGF). We regrettably and erroneously omitted the result of this statistical test in the initial submission of the manuscript. In addition to this, we also see that increasing concentrations of ligand (**Fig. 1B** and **Fig. 7C**) triggers a decrease the fraction of confined ligand-bound EGFR. Hence, while total EGFR (detected via Fab-Cy3B) and ligand-bound EGFR (detected via EGF-Cy3B) are initially very distinct in terms of mobility beginning at low doses of EGF stimulation, these two populations appear to begin to converge as the dose of EGF used for stimulation is increased.

Importantly, the percentage of EGFR that is ligand bound, shown in **Figure S1F** is an estimate that is based on the assumption that a single K_d defines the binding of EGF to EGFR. There is considerable evidence that this may not be accurate to describe the binding of EGF to EGFR at high doses of EGF. For instance, some studies have proposed negative cooperativity of binding of EGF to EGFR dimers that occurs subsequent to binding of EGF to one of the subunits in the dimer. This would mean that it is difficult to estimate the fraction of EGFR bound to EGF at higher EGF concentrations, and further suggests that EGFR may be less occupied with ligand than predicted in the calculation in **Figure S1G** at 100 ng/mL or above. Consistent with this interpretation, Needham et al found that 100 ng/mL represented the peak of EGFR oligomerization, which requires the formation of an interface between two unliganded EGFR subunits in separate EGFR dimers (Needham SR, et al. Nat Commun. 2016. 7:13307. PMID: 27796308). In the same study, it was only at higher concentrations of EGF (200-1000 ng/mL) where EGF was present in sufficient concentration to suppress oligomerization, reflecting that it was only at this higher range of EGF (well over 200 ng/mL) at which more complete receptor occupancy with ligand was achieved. Hence, the calculation presented in **Figure S1G** may be useful at approximating EGFR ligand occupancy at lower EGF concentrations, but perhaps not at higher EGF concentrations. We have updated the manuscript to better reflect this point, including in the description of ligand occupancy (Results section, page 4).

This complexity of the regulation of EGF binding to EGFR further supports the main conclusions of our study, in which we reveal a novel mechanism in which ligand binding to EGFR is regulated by tetraspanins. However, this complexity of binding of EGF to EGFR, including the negative cooperativity of EGF binding to the second subunit of a dimer, makes direct comparisons at high

dose EGF stimulation of the mobility of ligand-bound EGFR (detected via EGF-Cy3B) and total receptors (detected via EGF-Cy3B in the presence of unlabelled EGF) difficult to interpret. When EGFR is stimulated with higher doses of EGF, a larger proportion of EGFR binds EGF and undergoes internalization than when cells are stimulated with low doses of EGF. This results in loss of both EGFR and CD81 from the cell surface (See **Response Figure 1** below), making study of EGFR remaining at the cell surface under conditions with high dose EGFR difficult to interpret. Of note, stimulation of cells with lower doses of EGF as used throughout the study (10 ng/mL EGF) does not appreciably change the cell surface levels of EGFR or CD81 (See below). Moreover, we also observe that stimulation of EGFR with high doses of EGF (400 ng/mL, per Needham et al 2016 required to approach complete ligand occupancy) leads to apparent remodeling of clathrin structures at the cell surface, as seen by fewer and less prominent eGFP-clathrin structures observed by TIRF microscopy (**Response Figure 1** below).

Response Figure 1. ARPE-19 cells stably expressing eGFP-clathrin were labeled with Fab-Cy3B (to label total EGFR) and stimulated with EGF as indicated, followed by fixation and staining with CD81 antibodies. (A) Shown are representative images obtained by TIRF-M; antibody labeling of tetraspanins is highly specific (Fig. S2); similar experiments with labeling of CD82 and CD151 were performed (Fig. S4). Scale = 5 μ m.

To address the concern regarding the use of an additional fluorophore to validate that the Fab-labelling strategy did not alter EGFR behaviour in a relevant manner, we developed a complementary strategy for assessing EGFR mobility by generating cells that stably express a N-terminal SNAP tag within the EGFR coding region. This allows us to study EGFR mobility using an alternative labelling strategy and supports comparison at lower (10 ng/mL) EGF concentrations. Further to this being a separate labelling strategy, we generated stable cells using the doxycycline-inducible Sleeping Beauty transposon system, which allowed us to express N-SNAP EGFR at near-endogenous levels so as to not alter EGFR behaviour by overexpression (**Fig. S1F**), as we have done previously for expression of other proteins (Cabral-Dias, J Cell Biol. 2022 221:e201808181. PMID: 35238864). This strategy also allowed us to selectively label membrane-

bound EGFR using the cell-impermeant SNAP-Surface-488 (SNAP-488) reagent. Importantly, this system allows us to measure ligand-bound and total EGFR mobility using two separate fluorophores in the same cell, eliminating the possibility that cell-to-cell heterogeneity contributed to these outcomes.

To directly assess whether each fluorophore and labeling strategy may reveal distinct EGFR mobilities, we compared mobility in cells labeled with either EGF-Cy3B alongside SNAP-488 (EGF-Cy3B + SNAP-488) or Fab-Cy3b alongside SNAP-488 (Fab-Cy3B + SNAP-488). Our results show that SNAP-488-labeled EGFR exhibits a pattern of EGFR mobility that very closely resembles that measured with Fab-Cy3B, i.e. the majority of receptors exhibit mobile behaviour (shown in new panels, **Fig. 1E-H**). In contrast, in cells labeled with EGF-Cy3B + SNAP-488, there is a stark difference in the fraction of mobile receptors when comparing total receptors (via SNAP-488 labeling) and ligand-bound receptors (via Cy3B-EGF) (**Fig. 1 E-H**). Together, these results support our view that our strategies to track EGFR mobility are minimally perturbing. In addition to the new data shown in **Fig. 1 E-H**, we have updated the Results section (page 4) and Methods section (page 13) of the manuscript to reflect these results.

Furthermore, the experiments conducted with erlotinib reveal that in cells treated with this inhibitor, the mobility of total EGFR (detected via Fab-Cy3) and ligand-bound EGFR (detected via EGF-Cy3B) are nearly identical (**Fig. 8**). This provides yet further support for the conclusion that the differences in mobility of ligand-bound and non-ligand-bound EGFR (e.g. Fig. 2) reflect differences in the behaviour of these receptor classes and not artificial differences resulting from label-based alterations in EGFR mobility. We updated the manuscript in the Results section on page 10 to more explicitly make this point.

Together, these new results with the SNAP labelling of EGFR as a complementary method to support single particle tracking of the total receptor pool have allowed us to strengthen our interpretation that unliganded EGFR exhibits distinct mobility from ligand-bound EGFR. Some additional modifications to the text have also allowed us to strengthen the manuscript by clarifying ligand binding regulation.

2. In Figure 1C, the diffusion coefficient of the mobile fraction is stated to significantly increase with EGF treatment (from about 0.4 to above 0.5, as estimated from the plot). The authors say that this suggests that EGF stimulation releases some receptors from small scale interactions that restrict diffusion. However, this increase of diffusion coefficient of the mobile fraction upon treatment with EGF, as measured by Fab-Cy3B, is not detected in Figure 3 (3D vs 3E, control siRNA) or Figure 5 (5D vs 5E, control siRNA). How do the authors reconcile these differences?

We thank the reviewer for noting this. We do see a significant but small increase in the diffusion coefficient of EGFR (tracked via Fab-Cy3B) when stimulated with ligand in the results reported in **Figure 1C**. We do not directly compare the effect of EGF stimulation on the diffusion coefficient of EGFR in basal versus EGF stimulated conditions (detected via Fab-Cy3B) in the figures for the rest of the manuscript. Nonetheless, it does not appear that there is a notable difference in the diffusion coefficient of the mobile fraction of EGFR detected via Fab-Cy3B when comparing basal

and EGF-stimulated conditions. Hence, while this change in behaviour of the mobile fraction of total EGFR (detected via Fab-Cy3B) may reflect changes in small-scale interactions of EGFR upon ligand binding, this does not appear to be a robust manifestation of changes in EGFR protein-protein interaction. Our new results using SNAP-tagged EGFR as the mode of detection and tracking of total EGFR supports this, in that the diffusion coefficient of mobile EGFR (tracked via SNAP-488) does not appear to change upon stimulation with EGF (new **Fig. 1G**). Including this new data from SNAP-A488 tracking of EGFR has allowed us to temper the interpretation of this result (page 4).

3. In Figure 1, scale should be “3 μm ” not “3 μM ”.

We thank the Reviewer for pointing out this error and have updated the manuscript accordingly.

4. In Figure 2, is the CD81 signal constant during the 15 min of EGF treatment? (It appears to decrease with time in the images shown in Figure 2A).

We thank the Reviewer for this comment and have performed additional analyses on these images to quantify the abundance of CD81 on the cell surface (**Fig. S3H**). We see no difference in the total fluorescence intensity of CD81 within TIRF images. In addition to the new **Fig. S3H**, we have updated the manuscript on page 5 to reflect these results.

5. In Figure 2, is there ligand induced receptor endocytosis during the time scale of these experiments? In other words, is the total number of receptors at the membrane constant? The results in Figure 2 are interpreted as suggesting that there are 2 scales of association (line 206), of EGFR and tetraspanins, which is an over-interpretation of the results.

We thank the Reviewer for this comment and the opportunity to address how endocytosis may contribute to our results. As we see it, there are two aspects to this question:

1. First is that of the effect of EGF stimulation on the levels of EGFR at the cell surface, as large changes in the levels cell surface EGFR among conditions may impact the apparent mobility of the remaining cell surface EGFRs. Under the conditions that we examined, cells treated with lower doses of EGF (~10 ng/mL at which we conducted the majority of our experiments), there does not appear to be a large difference in the levels of EGFR at the cell surface, as detected by Fab-Cy3B, under the conditions examined (**Response Figure 1**). Many EGFR molecules will eventually undergo internalization in the ligand-bound state. However, under the conditions examined, the fraction of EGFR that undergoes endocytosis within the timeframe of labelling and measurement is small and any differences in EGFR mobility between stimulation and basal conditions are unlikely to be indirectly caused by substantial differences in cell surface EGFR.
2. Second is the possibility that the silencing of clathrin may impact EGFR mobility as an indirect effect of perturbation of endocytosis. To address this possibility, we examined the effect of silencing of dynamin2 on EGFR mobility (shown in a new **Fig. S9B**). We previously showed that silencing dynamin2 impaired EGFR internalization but not the recruitment of EGFR to clathrin structures (Garay et al. Mol Biol Cell 2015. 26:3504-19).

PMID: 26246598), likely as dynamin2 prevents clathrin-coated pit scission from the plasma membrane but does not prevent the formation of clathrin structures or the recruitment of EGFR therein. These experiments reveal that silencing dynamin2 had no effect on EGFR mobility (**Fig. S9B**). In contrast, silencing clathrin led to a loss of confinement of ligand-bound EGFR but not total EGFR in the absence of ligand (**Fig. 5**). Since both clathrin and dynamin 2 silencing led to similar loss of EGFR endocytosis in ARPE-19 cells (Garay et al. Mol Biol Cell 2015. 26:3504-19. PMID: 26246598), the effect of clathrin silencing on EGFR mobility is unlikely due to a perturbation of endocytosis, but instead reflects perturbation of the confinement of EGFR within clathrin structures at the plasma membrane. Moreover, as dynamin2 siRNA blocks EGFR endocytosis, these results also show that any modest loss of EGFR abundance at the cell surface upon stimulation with low doses (10 ng/mL EGF) does not change the apparent mobility of EGFR.

We have also included additional experiments and analyses to examine how EGFR may associate with tetraspanins at multiple levels: small-order assemblies and larger assemblies that may represent bona-fide tetraspanin nanodomains. We conducted an additional analysis to provide further clarity for the reader (**Fig. S3I-J**, size-based analysis) and have modified the Results section on pages 5-6 accordingly. These results show that in images acquired by TIRF microscopy, CD81 structures are found to exhibit a range of intensities, and as these are diffraction-limited objects, this indicates that tetraspanins are found in a range of sizes of structures at the plasma membrane (**Fig. S3I**). This is consistent with our interpretation that EGFR has at least 2 scales of association with tetraspanins. We further extend our analysis of tetraspanin objects and show that classification of tetraspanin objects into cohorts based on tetraspanin intensity indicates that both “large” and “small” tetraspanin objects associate significantly with EGFR (**Fig. S3J**). This further provides evidence to support our interpretation of EGFR association with at least two scales of tetraspanin assembly.

These measurements of CD81 assembly are based on intensity, since CD81 are diffraction-limited objects. To further extend our analysis of tetraspanin structures at the cell surface, we performed STED microscopy to image CD81, which allows us to observe CD81 structures at higher resolution and thus base observation of tetraspanin object size on more than fluorescence intensity. These images show that CD81 is observed in large assemblies consistent with tetraspanin nanodomains, as well as a range of smaller objects (new **Fig. S3K**). Together, these results, based on 3 complementary analyses that include the effect of tetraspanin silencing on EGFR mobility by single particle analysis, size analysis of CD81 objects in TIRF images, and STED images of CD81, support our interpretation that there are different scales of EGFR association with tetraspanins. We nonetheless agree that our results do not support that there exist exactly two discrete orders of tetraspanin assembly with which EGFR associates and rather a range of sizes of CD81 assemblies that likely range from small-scale dimers to larger nanodomains. We have updated the manuscript to reflect this, but this does not change the major aspects of our conclusions based on comparing the mobility of unliganded and ligand-bound EGFR.

6. In Figure 3, the same data on “movement class” are presented twice in Figures 3A and 3B (replotted), which is unnecessary.

We thank the Reviewer for this feedback. We showed this figure plotted this way as we have shown other analogous figures in a similar way. We agree that the effect of CD81 silencing on mobility of unliganded EGFR is robust and evident without re-plotting the immobile/confined cohorts in Fig. 3A-B. However, in some of the conditions where there is no significant effect on the fraction of confined/immobile EGFR, showing the replotted data may allow the reader to be able to appreciate differences (or lack thereof). For consistency, we would like to show the data in Fig. 3A-B with the data replotted, but we have added a statement in the figure caption to indicate which data is replotted, to avoid confusion.

7. Is it possible that the effects observed upon treatment with of clathrin siRNA are partly due to inhibition of endocytosis?

We thank the reviewer for this comment and have performed additional experiments to determine whether the effects of clathrin siRNA are due to inhibition of endocytosis. Briefly, we measured liganded or total EGFR mobility in cells depleted of dynamin2, one of the core proteins of the clathrin-mediated endocytic machinery; depletion of dynamin2 inhibits clathrin-mediated endocytosis but does not prevent clathrin structures from forming at the cell surface and EGFR from being recruited therein (Garay et al. Mol Biol Cell 2015. 26:3504-19. PMID: 26246598). Our results show that silencing of Dynamin-2 is without effect on the mobility of liganded or total EGFR (**Fig. S9B**). This suggests that the changes in EGFR dynamics that we observe following clathrin siRNA are not due to inhibition of endocytosis. Instead, the selective loss of EGFR confinement for the ligand-bound receptor upon clathrin silencing is due to loss of clathrin-coated pits capable of “capturing” the ligand-bound EGFR. In addition to this new **Fig. S9B**, we added a text in the Results on page 7 to discuss this experiment.

Reviewer #2 (Remarks to the Author):

Sugiyama et al. report results on using single-particle tracking (SPT) to study spatial regulation of EGFR by a set of tetraspanins, namely CD81, CD82, CD9, and CD151. By labeling ‘total’ EGFR with an anti-EGFR Fab and ‘ligand-bound’ EGFR with EGF, both of conjugated to Cy3B, they studied the diffusion properties of both EGFR populations under both basal conditions and various perturbations. They found that, whereas EGFR (total) is mostly (~80%) mobile, ligand-bound EGFR is much more likely (~50%) in the confined or immobile states (Figure 1). Upon ligand (EGF) stimulation, the amount of EGFR associated the three tetraspanins remained unchanged, so did the amount of CD81 associated with EGFR, but the amount of CD82 and CD151 associated with EGFR decreased significantly (Figure 2). CD81 knock down with siRNA led to a higher fraction of total EGFR in the mobile state (and concomitantly lower fractions in the confined or immobile states) – with or without EGF stimulation, where the mobile EGFR molecules also exhibited faster diffusion, but it had little effect on the diffusion properties of ligand-bound EGFR

(Figure 3). Knocking down the tetraspanins (particularly CD81) decreased the fraction EGFR capable of ligand-binding but increased the fraction that binds mAb806, an EGFR antibody with ectodomain conformation selectivity (Figure 4). Interestingly, tetraspanin knockdown had little effect on common tyrosine phosphorylations on EGFR (with the exception of pY992) or on ERK or AKT activation (Figure 7). Blockade of tyrosine phosphorylation of EGFR using TKIs such as erlotinib increased the mobile population of ligand-bound EGFR as well as the mobility of this population but had little effect on the diffusion of total EGFR (Figure 8). Combining these results and modeling (Figure 6), other evidence not thoroughly summarized here (Figure 5 on clathrin), the authors suggest a model where EGF preferentially binds to EGFR pre-confined in tetraspanin nanodomains and, upon EGF stimulation, leaves these nanodomains but quickly become associated with clathrin-based nanodomains.

Regulation of EGFR by tetraspanins has been studied extensively but, as the authors rightfully put, gaps remain in our understanding of the exact roles of tetraspanins in the complex process of EGFR ligand binding, activation, and signal propagation. Overall, the study is helpful in addressing these gaps, and the use of SPT and subsequent SPT data analysis are well appreciated. The problem with this work, however, is that the authors try to describe the entire process – from baseline EGFR diffusion and confinement to EGF binding selectivity to EGFR translocation (presumably from one nanodomain to another) to signal propagation in a single study essentially with SPT alone. This is a very ambitious undertaking. I am afraid that, even with the large amount of data presented and the great length that the authors have taken to try to reconcile all the observations, most conclusions are ambiguous rather than definitive. After reading the manuscript many times I still could not put all the pieces together. In my opinion, it would have been much better if the authors focused on a single point (such as the role of tetraspanins in ligand gating) and bring it to a close. Below are some of the issues that I had.

We thank the Reviewer for their thorough review of the manuscript and for the positive comments regarding the scope of the study. We appreciate that the Reviewer has identified the role of tetraspanins in the control of EGFR ligand binding as the core thrust of the manuscript. As we discussed in the introduction, our SPT approach built upon previous studies of EGFR membrane dynamics with respect to nanodomains that often relied on harsh perturbations (e.g. cholesterol disruption or actin depolymerization) or SPT methods that confounded the results/interpretations. We could not improve upon these previous studies without developing a novel suite of tools and using them to systematically characterize EGFR confinement/ligand-binding in tetraspanins and other membrane nanodomains. As we describe in the manuscript, EGFR confinement within tetraspanin nanodomains imparts upon EGFR the ability to bind ligand, which precedes transit of EGFR to clathrin nanodomains. Achieving this level of resolution, of single receptors, requires a SPT approach. Where possible, we have supplemented the SPT experiments with other methodologies to provide the reader with a more global view of how these nanoscale interactions affect broader cellular functions.

Below, we provide our responses to the Reviewer's specific comments and concerns.

First, on the imaging of total vs ligand-bound EGFR. The two populations were probed using Cy3B labeled Fab and EGF, respectively, it was not clear to me whether EGFR populations detected by the two probes overlap or exclude each other. Cy3B-Fab was stated to probe 'total EGFR' so my initial assumption was that EGFR detected by Cy3B-EGF should be a subset of that detected by Cy3B-Fab. However, as I read on I became less and less certain about this. For example, in Figure 1C and D, there was not any indication that a subpopulation of the 'total' EGFR corresponds to the ligand-bound EGFR (i.e., exhibiting similar diffusion properties). Same thing for Figure 3 E&F, and H&I. While this could be because the ligand-bound population is small, the authors estimated that 10-40% of total EGFR should be ligand-bound at 5-10 ng/mL EGF and 70-90% at 100 ng/mL. At these binding fractions, the diffusion properties (and the changes thereof) of the ligand-bound population should have been more clearly reflected in those of the total EGFR.

This is an important point and we have added new experimental evidence and additional comparisons to support the interpretation that the Cy3B-EGF is indeed a subset of Fab-Cy3B. It is worth noting that our original submission already provided evidence that Fab-Cy3B does not alter ligand binding, as we showed that saturating Fab labeling did not alter EGF-Cy3B binding (**Figure S1A-C**). With this result, it is difficult to imagine how Cy3B-EGF labeling would not reflect a subset of Fab-EGF labeling. Nonetheless, the point by the reviewer that we should ensure that this interpretation is supported by additional evidence and comparisons as suggested is important.

For ligand-bound EGFR (detected by labelling with EGF-Cy3B) to not represent a subset of total EGFR detected by Fab-Cy3B, the labelling of total EGFR with Fab-Cy3B would have to impair ligand binding. We had already included experiments that show that this is not the case (Figure S1A-C). To address this further, we developed an additional, orthogonal method to label total EGFR, suitable for single-particle tracking experiments. We generated an EGFR with an exofacial SNAP tag. Further to this being a separate labeling strategy, we also generate stable cells using the doxycycline-inducible Sleeping Beauty transposon system to ensure controlled expression level of SNAP-tagged EGFR at levels 2-3x endogenous (new panel **Figure S1F**).

Using this system we are able to express N-SNAP EGFR and selectively label plasma membrane EGFR using the cell-impermeant SNAP-Surface-488 (SNAP-488) reagent (**Fig. 1E-H**). Importantly, this system allows us to measure ligand-bound and total EGFR mobility using two separate fluorophores in the same cell, as we can obtain a 10s time-lapse of SNAP-A488 fluorescence followed by a 10s timelapse of EGF-Cy3B or Fab-Cy3B fluorescence from the same cell. To directly assess whether the fluorophores/labeling strategies impact EGFR differently, we compared mobility in cells labeled with either EGF-Cy3b + SNAP-488 or Fab-Cy3b + SNAP-488. Our results show that SNAP-488-labeled EGFR exhibits a pattern of EGFR mobility that closely resembles that measured with Fab-Cy3B, i.e. the majority of receptors exhibit mobile behaviour (**Fig. 1F**). In contrast, in cells labeled with EGF-Cy3b + SNAP-488, there is a stark difference in the fraction of mobile receptors when comparing total receptors (labelled using SNAP-488) and ligand-bound receptors (labelled using EGF-Cy3B) (**Fig. 1F**). Further, when examining the diffusion coefficient (**Fig. 1G**) and confinement radius (**Fig. 1H**), we see that the SNAP-488 and Fab-Cy3B labelling reveal similar parameters for the mobility of total EGFR that is clearly distinct

from that of ligand-bound EGFR detected via EGF-Cy3B. Together, these results strengthen the interpretation that EGF-Cy3B labels a subset of total EGFR labelled by either Fab-Cy3B or SNAP488. We have updated the manuscript in the Results section on page 4 and the Methods section on page 13 to describe these new experiments shown in **Fig. 1E-H** and **Fig. S1F**.

Second, on the preferential binding of EGF to EGFR pre-confined in the tetraspanin nanodomains. The main arguments for this conclusion include: (a) EGF stimulation led to decreased amount of tetraspanins associated with EGFR (but not the amount of EGFR associated with tetraspanin); (b) knock down of CD81 decreased the confined and immobile fractions of total EGFR regardless of EGF stimulation, but it did not change the mobility of EGF-bound EGFR; (c) the fraction of ligand-bound EGFR (relative to all probed by Cy3B-EGF) associated with CD82-positive membrane regions was greater than that of the total EGFR (probed by Cy3B-Fab) at time points as early as 15s after incubation with EGF (Cy3B-EGF); and (d) knocking down CD81 or all 4 tetraspanins reduced EGF binding to cell surface. These observations support, but do not prove, that EGF binding preferentially takes place in large tetraspanin nanodomains (where a subset of EGFR is confined).

We thank the Reviewer for raising this issue. Our results suggest that upon ligand binding, EGFR does not associate with larger tetraspanin nanodomains, since silencing tetraspanins does not affect the fraction of ligand-bound EGFR that is confined. As such, upon ligand-binding, we expect EGFR to rapidly dissociate from tetraspanin nanodomains. In our initial submission, we examined the potential preferential localization of EGF-Cy3B to tetraspanin nanodomains, finding that there is a preferential binding of EGF-Cy3B to tetraspanin nanodomains compared to the labelling with total EGFR (detected via Fab-Cy3B). Despite this being an early timepoint of EGF stimulation, we expect that the labelling of EGF-Cy3B within tetraspanin nanodomains may under-represent the fraction of EGFR that initially bound within tetraspanin nanodomains, as binding of EGF to EGFR should lead to rapid loss of EGFR from tetraspanin nanodomains, and thus not all ligand-bound EGFR will be “captured” at the site of ligand binding in the experiment. Technical limitations preclude us from measuring EGF binding earlier than 15s, and in tracking both tetraspanin nanodomains and these early timepoints of EGF binding simultaneously. To address the reviewer’s point, we have made edits throughout the manuscript to ensure that we do not over-interpret these results and state that EGFR binding occurs preferentially (but perhaps not exclusively) within tetraspanin structures (of various sizes), and that this may include binding within larger tetraspanin nanodomains that establish confinement of EGFR.

Despite (c), the fractions of total or ligand-bound EGFR found on CD82-positive spots were only 10-15% at time points as early as 15s. According to the simulation model presented in Figure 6, the half time for EGFR to exit these tetraspanin domains are 83 ± 8 s. Does that mean that the majority of ligand-binding of EGFR (>80%) took place in CD82-negative regions? Additionally, whereas the reduced EGF binding capacity of EGFR after tetraspanin knockdown does suggest a role of tetraspanins in regulating ligand binding (which had been well established), there was no other direct evidence presented to directly prove that the loss of EGF binding was caused by reduced confinement in tetraspanin domains. In Figure 4B, ligand binding was lost by (only) 30-

40% when all 4x tetraspanins were knocked down; would this further suggest that tetraspanin domains do not necessarily 'dominate' ligand binding of EGFR?

We thank the Reviewer for the opportunity to clarify this point, which we realize was not clearly described in our initial submission. In the computational model of EGF binding, the energy barriers confining EGFR to tetraspanin nanodomains are immediately (0s) removed after ligand binding. Through diffusion, EGFR then rapidly exits the tetraspanin nanodomain. Effectively, ligand binding promptly removes an EGFR from a tetraspanin nanodomain. The rationale for this is 1) the lack of effect of tetraspanin silencing on the fraction of EGFR that is immobile/confined for ligand-bound EGFR (**Figure 3C**) and 2) the loss of EGFR from association with tetraspanin nanodomains following EGF stimulation (**Figure 2**). The computational model shows that we can recapitulate some of the major experimental observations of the study, namely, that EGF stimulation in the range of 10-25 ng/mL EGF is largely without effect on the fraction of total EGFR that is immobile/confined, with a sufficiently slow rate of EGFR entry into tetraspanin nanodomains (83 +/- 8s). In this model, the parameters are simplified so that ligand binding can only occur within tetraspanin nanodomains. We appreciate that this computational model simplifies the binding of EGF to EGFR, such that this binding is only permitted within tetraspanin nanodomains in the model and our data reflect a preference for (but perhaps not exclusive) EGFR ligand binding within tetraspanin nanodomains. However, this modelling approach does allow us to resolve that the rate of entry of EGFR into tetraspanin nanodomains to ensure confinement is relatively slow. We have updated the manuscript in the results section, page 8, to more clearly describe the parameters of the model, including the EGFR exit times from tetraspanin nanodomains when ligand-bound.

While we appreciate that the Reviewer highlighted that our results with respect to loss of tetraspanins might explain only 30-40% of total EGFR membrane binding, we have taken additional care to highlight two major findings: 1) that tetraspanins selectively control confinement of a large subset of total surface EGFR, and 2) that tetraspanins control ligand binding of a subset of total surface EGFR. As we discuss in the introduction and discussion sections, these findings help resolve one of the longstanding issues in understanding the regulation of EGFR ligand binding, namely the observation that only a small percentage of EGFR is capable of binding ligand.

While previous studies indicate that tetraspanins regulate EGFR signaling in some way, these studies do not identify the mechanism or stage at which tetraspanins control EGFR signaling. For example, a previous study showed that CD82 overexpression resulted in a reduction in EGFR dimer formation upon stimulation with EGF and other ligands (Odintsova E, et al. 2003. 116:4557-66. PMID: 14576349). This important study was key to linking tetraspanins to regulation of EGFR signaling, but this study did not show that regulation of EGFR signaling by CD82 occurred due to changes in ligand binding. In addition, the strategy of overexpression of CD82 may have broader effects than disruption of EGFR interaction with tetraspanin nanodomains. Other studies that used siRNA gene silencing (e.g. of CD82) showed disruption of EGFR signaling (e.g. to. Erk) but did not resolve the mechanism of regulation of EGFR signaling by CD82 (Danglot L, et al. J Cell Sci. 2010. 123:723-35, PMID: 20144992). Hence, our work is novel in that we reveal both preferential

binding of EGF to EGFR within tetraspanin nanodomains as well as loss of EGF binding upon tetraspanin silencing. Our work also shows the distinct contribution of tetraspanins to confinement and mobility of EGFR in the basal and ligand-bound states.

Hence, to the best of our knowledge, this is the first study to show that tetraspanins regulate EGFR ligand binding, thus representing an important novel observation. We agree that resolving the distinct functional (e.g. signaling) behaviour of EGFR by mobility classes is one of the major challenges at the leading edge of receptor signaling, as studied using single particle tracking methods. This is what led us to expand upon single-particle tracking, to examine the functional role of tetraspanins for EGF ligand binding (**Fig. 4B**) and the preferential association of EGF-Cy3B with tetraspanins structures (**Fig. 4A**). With the additional evidence that we provide that supports the interpretation that EGF-Cy3B labels a subset of total EGFR (as discussed above), the effects of tetraspanin (i.e. CD81) perturbation on EGFR single-particle tracking better support the interpretation that EGFR confined within tetraspanin nanodomains exhibits preferential ligand binding. Specifically, we indicate that our novel findings that tetraspanins, and in particular CD81, regulates both EGFR ligand binding and nanodomain confinement. We further indicate that the complementary experiments involving comparison of ligand-bound and total EGFR mobility, and preferential binding of EGF-Cy3B in CD81 suggests that EGF binding occurs preferentially with EGFR confined within tetraspanin nanodomains. To address this point, we have made edits to the Discussion to highlight how our work significantly expands on prior knowledge of EGFR regulation by tetraspanins (Page 11)

Third, on the explanation for the lack of effect on EGFR phosphorylation (except pY992) or signaling upon tetraspanin silencing despite significant reduction in ligand binding. The authors suggested that this is likely due to compensation through increased EGFR oligomerization, which 'amplifies' the effect of low ligand binding by cross phosphorylation of receptors in the same oligomer. Again, while this is plausible, there was not enough evidence to prove that this is indeed the case. How EGFR oligomerization compensates the loss in dimerization and subsequent intra-dimer phosphorylation to yield nearly identical receptor phosphorylation and ERK/AKT signaling is unclear. Besides, tetraspanin silencing reduced the confinement of total EGFR (Figure 2) - should that have reduced oligomerization instead?

We thank the reviewer for the opportunity to clarify this point. The measurement of phosphorylation of Y992 was established by a previous study (Needham SR, et al. Nat Commun. 2016. 7:13307. PMID: 27796308) as requiring EGFR oligomerization. As such, it is the best available direct marker of the extent of EGFR oligomerization. This allows us to conclude two things:

- 1) In control cells, there is EGF-stimulated Y992 phosphorylation, which indicates that EGFR oligomerization does occur. Furthermore, since EGFR oligomers form between ligand-bound EGFR and unliganded EGFR dimers (Needham SR, et al. Nat Commun. 2016. 7:13307. PMID: 27796308), increasing the concentration of EGF (100 ng/mL) is expected to reduce EGFR oligomerization. This is indeed what we observe, as measured by a decrease in the fraction of confined/immobile EGFR at 100 ng/mL EGF compared to 10 ng/mL EGF stimulation.

- 2) Upon CD81 silencing, we see a robust increase in pY992 on EGFR, which supports the interpretation that CD81 silencing suppresses oligomerization between ligand-bound EGFR and non-ligand bound EGFR, as reported by Needham et al. The EGF-stimulated phosphorylation of PLCg1 is dependent on phosphorylation of EGFR on Y992. We have now added further results showing that CD81 silencing also significantly and robustly increases PLCg1 phosphorylation (**Fig. 7F**), further supporting this point.

Taken together, our data indicates that CD81 silencing leads to fewer EGFR dimers bound to ligand (as measured by reduced EGF-Cy3B binding, **Fig. 4B**), and increased oligomerization based on the selective enhancement of pY992 phosphorylation (**Fig. 7F**). As shown in the Needham et al. study, EGFR oligomers can be detected with as little as 4 EGFRs (2 dimers in sequence) or >12 (6 dimers in sequence). Our data thus suggests that CD81 silencing may cause a shift towards larger oligomers (~12 EGFRs or more), while smaller oligomers (~4-6) predominate in the control EGF-stimulated condition.

Furthermore, EGFR oligomerization is expected to be intimately tied to recruitment to clathrin coated pits, and these two parameters are difficult to separate with single particle tracking approaches. Previous studies have noted that clustering of internalization-capable receptors (e.g. transferrin receptor, which is constitutively internalized) leads to enhanced initiation of clathrin-coated pits containing these clustered receptors due to avidity effects of clathrin adaptor proteins binding to their cytoplasmic binding sites on the clustered receptors (Liu AP et al J Cell Biol. PMID: 21187331). Hence, EGFR oligomers are likely to exhibit high degrees of confinement as a result of both the size of the oligomer, as well as the concomitant capture of oligomers (i.e. receptor clusters) by nascent clathrin coated pits structures. Consistent with this, we do see a partial loss of confinement of EGFR by clathrin silencing, selectively in the ligand-bound state (Fig. 5C). However, because the confinement of ligand-bound EGFR reflects the confinement imposed by oligomer size and clathrin-coated pit capture, it is not possible to compare the mobility of ligand-bound EGFR (via tracking of EGF-Cy3B) in control versus CD81-silenced cells to resolve a role for CD81 in suppressing EGFR oligomerization. We have added a statement in the Discussion on page 11 to clarify this point.

With regard to the specific question about how tetraspanin silencing impacts confinement of total EGFR, we do indeed see that in the absence of ligand, CD81 silencing reduces the confinement of total EGFR (**Fig. 3A**). We also see that CD81 silencing reduces the confinement of total EGFR when cells were stimulated with EGF (**Fig. 3B**); however, it is worth noting that in this condition, total EGFR is a mixture of ligand bound and non-ligand bound EGFR. In contrast, the mobility of just selectively the ligand-bound EGFR (tracked via EGF-Cy3B labeling) is unaffected by CD81 silencing (**Fig. 3C**), supporting that only basal EGFR is directly subject to confinement by CD81. This is consistent with our model that CD81 and other tetraspanins may limit the ability of non-ligand-bound EGFR to engage with ligand-bound EGFR to lead to oligomer formation, such that loss of CD81 leads to increased oligomer-dependent EGFR pY992 signaling. To address this point, we have added a brief explanation of this in the Discussion section, on page 11.

Fourth, regarding the modeling, which took place before the discussions on oligomers. As a result, the modeling did not take into account receptor oligomerization, making the value of this effort somewhat questionable. While modeling can be very useful in this work, it does require a relatively clear understanding of the critical processes, which is not yet the case. Among other things, tetraspanin domains may not be the only membrane structures regulating diffusion and ligand-binding of unliganded EGFR (this should be taken into account in data interpretation throughout the manuscript, not just the modeling).

We thank the reviewer for raising this point and providing an opportunity to clarify the contribution of the modeling in the manuscript. As discussed above, EGFR oligomerization is expected to be swiftly followed by association with nascent clathrin-coated pits. This is motivated by similar behaviour of clathrin capture of oligomeric transferrin receptors (Liu AP et al J Cell Biol. PMID: 21187331). To simplify the quantitative model and its analysis, we have not explicitly included EGFR oligomerization processes, instead focusing on diffusive search for and confinement within tetraspanin and clathrin domains. With respect to the effect on mobility of oligomerization and clathrin structures, a key experimental measurement is in Fig 1B, which shows that the non-mobile fraction of ligand-bound EGFR changes little with [EGF], with observations later in the manuscript suggesting this confinement of ligand-bound EGFR is largely to clathrin domains. The clathrin domain parameters were adjusted to provide this degree of mobility, and adding oligomerization to the model would produce a more complicated model that would in turn be adjusted to match this experimental mobility data. Furthermore, as discussed above, it is difficult to resolve the unique contribution of EGFR oligomerization from clathrin structures in confinement of ligand-bound EGFR, given that oligomerized EGFR is expected to be rapidly engaged by clathrin structures given the avidity effect resulting from clustering of clathrin adaptor (AP2) binding motifs on receptor cytosolic regions.

The key contribution of the modeling in this manuscript is to determine the quantitative parameters at which the experimentally-observed features occur. The correct ligand-bound EGFR mobility behaviour is important to determining the contribution of EGFR entry into and exit from tetraspanin domains to the mobility of all EGFR (both ligand free and ligand bound). Our modeling demonstrated that slow recruitment of EGFR to tetraspanin nanodomains is required to recapitulate the mobility of all EGFR shown in Fig. 1B, which shows that EGFR mobility changes little with [EGF]. We see value in this modeling outcome as it would be difficult to obtain similar estimates solely through experimental measurements.

In future studies, we plan to further resolve how EGF-stimulated EGFR oligomerization is intimately and reciprocally regulated by association with plasma membrane clathrin structures. This will require the development of experimental manipulations that can dissociate EGFR oligomerization formation from clathrin engagement, such as the use of specific EGFR mutants that disrupt face-to-face dimerization required for oligomer formation, as well as perhaps EGFR mutants with disrupted endocytic motifs. Study of such mutants would require knockout-rescue approaches with controlled levels of expression of EGFR constructs. As noted by the reviewer in a comment above, this study is already quite comprehensive in its undertaking of studying regulation of EGFR mobility and ligand binding at the cell surface. Since the oligomerization and

clathrin association is subsequent to ligand binding, we believe that resolving the regulation of oligomerization and clathrin association through new experiments and modelling approaches is beyond the scope of the study, but will be a focus of our future work.

Nonetheless, the reviewer raises an important point that there may be other mechanisms or structures than tetraspanin nanodomains, clathrin structures and receptor oligomerization that may regulate EGFR mobility either in the basal or ligand-bound state. We have added a paragraph in the Discussion section on page 11 to elaborate on this point.

Lastly, there were also some inconsistencies in the mobility analyses. In Figure 1, 'total' EGFR diffusing at $\sim 0.5 \mu\text{m}^2/\text{s}$, $\sim 0.2 \mu\text{m}^2/\text{s}$, and $\sim 0.1 \mu\text{m}^2/\text{s}$ was assigned 'mobile', 'confined', and 'immobile', respectively; by contrast, ligand-bound EGFR diffusing at $\sim 0.3 \mu\text{m}^2/\text{s}$, $\sim 0.13 \mu\text{m}^2/\text{s}$, and $\sim 0.11 \mu\text{m}^2/\text{s}$ (read from the plot directly so these numbers are approximate) was assigned the same three states. The 'confined' state of total EGFR has a diffusion rate almost as fast as the 'mobile' state of ligand-bound EGFR. What was the criterion for assigning these states as mobile, confined, or immobile? One would also need to be careful to call a state 'confined' simply based on a small diffusion coefficient; it could be slow but free instead of confined diffusion. Moreover, the tetraspanin domains were supposedly 100-200 nm, yet the 'confinement' radii for all the confined or even the immobile states were on the order 1-3 μm . Could the authors explain the discrepancies?

We have provided some additional details for the method used to analyse EGFR single particle image series. The classification of EGFR into mobility cohorts and the diffusion coefficients and confinement radii were determined using moment-scaling spectrum analysis, previously established in several studies. This more explicit description has been added to the method for this section, page 15.

The differences in the diffusion coefficients of ligand-bound versus unliganded EGFR (for example, in Fig. 1C, but in other figure as indicated by the reviewer) likely reflect small order assemblies of EGFR, which can include dimerization of the receptor that is induced by ligand binding. As expected, the diffusion coefficients of the immobile fractions are similar when comparing ligand-bound and total EGFR. The diffusion coefficient of the confined and especially the mobile fractions of ligand-bound and total EGFR are indeed different, with the ligand-bound receptors exhibiting a lower diffusion coefficient.

This lower diffusion coefficient of the liganded receptors likely reflects the ability of the ligand-bound receptor to engage in asymmetric dimers, with the resulting dimer and associated protein(s) of the ligand-bound receptor exhibiting lower diffusion than that of the unliganded receptors. That we do not see a change in diffusion coefficient when comparing total EGFR in the basal and EGF-stimulated condition may reflect complex changes in EGFR association with other proteins in small-scale assemblies, and that ligand-bound EGFR is a relatively small fraction of total EGFR at the concentration of EGF used in the majority of experiments in this study (at 10

ng/mL). We have added an explicit description of this in the Results section (Page 4), discussing the difference in diffusion coefficient of the ligand-bound and total EGFR.

Moreover, the treatment with erlotinib causes a sharp rise in the diffusion coefficient of the mobile and confined fractions of the ligand-bound receptor population (Fig. 8G), such that these diffusion coefficients are similar in this state to that of total EGFR (e.g. unliganded EGFR, Fig. 8E). This additional insight suggests that the differences in the diffusion coefficients in unliganded versus ligand-bound EGFR are not the consequence of artifacts of labelling, but reflect real differences (e.g. in small order assembly of proteins with mobile EGFR) that are disrupted upon erlotinib treatment.

With regard to the confinement radii of the immobile and confined fractions, these do appear larger than the radii of tetraspanin nanodomains (100-200 nm), and also larger than clathrin structures (also ~100-200nm). That the confinement radius of EGFR in these various conditions is larger than what might be expected from the size of nanodomains in which they are confined can be explained by limitations imposed by resolution of the imaging system and EMCCD camera used in these experiments. In addition, the single particle tracking analysis used assigns an entire track of EGFR to either immobile, confined, or mobile classes, and as such tracks that undergo transition (e.g. from mobile to confined/immobile) during a track may contribute to a larger confinement radius if this track is classified as confined/immobile. We have updated the manuscript to include a brief discussion of the observed confinement radius relative to the known size of tetraspanin nanodomains, on page 6.

REVIEWERS' COMMENTS

Reviewer #1 (Remarks to the Author):

The revised version of the manuscript includes new experiments to validate the use of SPT to investigate the mobility of total EGFR and ligand-bound EGFR at the cell membrane. In Figure 1, a cell line that expresses EGFR with a SNAP-tag allowed for detection of mobility of total EGFR and ligand-bound EGFR in the same cells. The central observation in the manuscript that ligand-bound EGFR shows lower mobility compared to total EGFR is further supported by these experiments.

In Figure 2, the colocalization between EGFR with CD82 and CD151 is shown to decrease or remain unchanged upon treatment with EGF, depending on whether the colocalization analysis is performed based on intensity or position. The authors interpret these results to mean that EGFR remains associated with tetraspanins upon EGF binding but loses association with tetraspanin nanodomains. I don't have the expertise to assess whether these analysis are valid but considering the central role of this interpretation, I think they should be revised by an expert.

In Figure 1, the labels D,E,F,G, should be E,F,G,H.

In the figure legend of Figure 4, in line 2, instead of "... (to label total EGF)..." it should read "... (to label total EGFR)..."

Reviewer #2 (Remarks to the Author):

The authors have addressed my comments, and I appreciate the new experiments that they have performed in response to my comments as well as reviewer 1. I now suggest that the manuscript be accepted for publication.

Sugiyama et al Nature Communications – Response to Reviewer Comments

April 6th, 2023

We have responded to the reviewer comments in blue below.

Reviewer # 1(Remarks to the Author):

The revised version of the manuscript includes new experiments to validate the use of SPT to investigate the mobility of total EGFR and ligand-bound EGFR at the cell membrane. In Figure 1, a cell line that expresses EGFR with a SNAP-tag allowed for detection of mobility of total EGFR and ligand-bound EGFR in the same cells. The central observation in the manuscript that ligand-bound EGFR shows lower mobility compared to total EGFR is further supported by these experiments.

We thank the Reviewer for their evaluation of the additional experiments using the SNAP-tag EGFR.

In Figure 2, the colocalization between EGFR with CD82 and CD151 is shown to decrease or remain unchanged upon treatment with EGF, depending on whether the colocalization analysis is performed based on intensity or position. The authors interpret these results to mean that EGFR remains associated with tetraspanins upon EGF binding but loses association with tetraspanin nanodomains. I don't have the expertise to assess whether these analysis are valid but considering the central role of this interpretation, I think they should be revised by an expert.

We thank the Reviewer for their feedback. The position-based colocalization analysis presented in Figure 2 has been rigorously validated. The original description of the method (Ref 59) was peer-reviewed and published; the main thrust of this article was description of this method for 2-3 colour colocalization analysis. In addition, the original authors made the code publicly available from initial publication. To address the concerns of the Reviewer, we modified the text in the discussion section to reflect that our interpretation is one possible interpretation of the colocalization data.

In Figure 1, the labels D,E,F,G, should be E,F,G,H.

We thank the Reviewer for identifying this error. The figure labels have now been revised.

In the figure legend of Figure 4, in line 2, instead of "... (to label total EGF)..." it should read "... (to label total EGFR)..."

We thank the Reviewer for identifying this error. The correct text now appears in the Figure 4 legend.

Reviewer #2 (Remarks to the Author):

The authors have addressed my comments, and I appreciate the new experiments that they have performed in response to my comments as well as reviewer 1. I now suggest that the manuscript be accepted for publication.

We thank the Reviewer for their evaluation of the revised manuscript and for their initial critique.